# A proteogenomic signature of age-related macular degeneration in blood

Valur Emilsson [1,2✉], Elias F. Gudmundsson [1], Thorarinn Jonmundsson [1], Brynjolfur G. Jonsson[1], Michael Twarog[3], Valborg Gudmundsdottir[1,2], Zhiguang Li[4], Nancy Finkel[3], Stephen Poor [3], Xin Liu[3], Robert Esterberg[3], Yiyun Zhang[3], Sandra Jose[3], Chia-Ling Huang[3], Sha-Mei Liao[3], Joseph Loureiro[3], Qin Zhang[3], Cynthia L. Grosskreutz[3], Andrew A. Nguyen[3], Qian Huang[3], Barrett Leehy[3], Rebecca Pitts [3], Thor Aspelund [1], John R. Lamb[5], Fridbert Jonasson[2,6], Lenore J. Launer [4], Mary Frances Cotch [7], Lori L. Jennings [3], Vilmundur Gudnason [1,2] & Tony E. Walshe [3✉]

Age-related macular degeneration (AMD) is one of the most common causes of visual impairment in the elderly, with a complex and still poorly understood etiology. Whole-genome association studies have discovered 34 genomic regions associated with AMD. However, the genes and cognate proteins that mediate the risk, are largely unknown. In the current study, we integrate levels of 4782 human serum proteins with all genetic risk loci for AMD in a large population-based study of the elderly, revealing many proteins and pathways linked to the disease. Serum proteins are also found to reflect AMD severity independent of genetics and predict progression from early to advanced AMD after five years in this population. A two-sample Mendelian randomization study identifies several proteins that are causally related to the disease and are directionally consistent with the observational estimates. In this work, we present a robust and unique framework for elucidating the pathobiology of AMD.

[1] Icelandic Heart Association, Holtasmari 1, IS-201 Kopavogur, Iceland. [2] Faculty of Medicine, University of Iceland, 101 Reykjavik, Iceland. [3] Novartis Institutes for Biomedical Research, 22 Windsor Street, Cambridge, MA 02139, USA. [4] Laboratory of Epidemiology and Population Sciences, National Institute on Aging, Bethesda, MD, USA. [5] Novartis Institutes for Biomedical Research, 10675 John Jay Hopkins Drive, San Diego, CA 92121, USA. [6] Department of Ophthalmology, University Hospital, Reykjavik, Iceland. [7] Division of Epidemiology and Clinical Applications, National Eye Institute, National Institutes of Health, Bethesda, MD, USA. ✉email: valur@hjarta.is; tony.walshe@agios.com

AMD is a progressive late-onset disease that primarily affects the macular area of the retina and is a common cause of permanent loss of vision in the elderly population[1]. The clinical hallmark of early AMD is an accumulation of extracellular protein and lipid-containing deposits between the retinal pigment epithelium (RPE) and Bruch´s membrane, termed drusen. Advanced AMD can be either neovascular (nAMD) associated with blood vessel growth and leakage, or geographic atrophy (GA/dry AMD) characterized by patches of RPE cell and photoreceptor cell loss in the macula. Anti-VEGF therapy is highly effective in controlling the abnormal vessel growth and leakage in nAMD, however, the disease nevertheless progresses and by seven years, 98% of nAMD patients have atrophy[2]. Patients with GA have marked visual disability with relentless visual deterioration and progression to legal blindness. There are currently no approved therapies for GA or early AMD[3,4]. Elucidation of AMD pathobiology and identification of modifiable targets are critical for identifying clinically relevant biomarkers and designing therapeutics.

The first common genetic risk factor reported for AMD was the missense variant rs1061170 in *CFH* on chromosome 1q31.3[5–8]. Many genes encoding other proteins involved in the complement cascade reside within the 1q31.3 region, including complement factors H related to 1 to 5 (*CFHR1-5*). Two independent variants at 1q31.3, rs1061170 and intronic variant rs1410996, account for 17% of AMD risk[9]. In the most recent GWAS examining 16,144 patients with advanced AMD, 52 independent variants were found at 34 different genomic loci explaining 46.7% of the variability in AMD risk[10]. The risk variants with the largest difference between late AMD patients and healthy controls reside within the *CFH* genomic region and the *ARMS2/HTRA1* locus on chromosome 10q26, although for most variants, the effect size was small[10]. While some recognized AMD causal candidates are found at these genomic risk locations, the vast majority are yet to be identified.

Proteins are undeniably the key players in all life processes, with changes in their function and/or regulation influencing disease and well-being. As a result, changes in protein regulation and function, as well as their related networks, are most likely to mediate the genetic risk of complex diseases[11–14]. Serum proteins have the desired attributes required for a comprehensive and unified approach to measuring an individual's global molecular status, as they capture information across many tissues and show a direct link to disease-related molecular pathways and activities[11,12]. Furthermore, serum proteins participate in cross-tissue regulatory loops, and therefore tissue-specific disease progression emerges from an integration of local and systemic signals[12]. Recent developments in high-throughput measurements of thousands of proteins in a single sample have aided this work[11,15–17], and aptamer-based affinity methods in particular, have been a driver of recent discoveries[11,18–21]. In this study, the serum levels of 4782 proteins, encoded by 4137 human genes, measured in 5457 individuals from the prospective population-based AGES-RS cohort were examined for association to different stages of AMD, disease progression and prediction, and the extent to which they mediated the genetics of the disease.

## Results

**An examination of 4782 serum proteins for links to various stages of AMD.** Descriptive statistics on the appearance of AMD in the 5457 person AGES-RS cohort ages 67 years and older (mean age $76.6 \pm 5.6$ years; 57.3% female) are shown in Supplementary Data 1. Using sex and age-adjusted logistic regression analysis of 4782 proteins (4137 gene symbols) and two distinct definitions of early-stage AMD (see Methods section for details),

we discovered that 28 serum proteins were associated with different stages of AMD using a study-wide significance threshold (Fig. 1a–c, Table 1, Supplementary Fig. 1A-C, and Supplementary Data 2 and 3). This included 15 proteins associated with early-stage AMD identified using both Holliday et al.[22] and Jonasson et al.[23] classification criteria, with two proteins unique to Jonasson et al.[23] and 10 proteins unique to Holliday et al.[22] (Fig. 1a, Supplementary Fig. 1A, and Supplementary Data 2 and 3). The relationship between the quintiles of the 28 protein levels and AMD outcomes revealed that some were only associated with the early or late stages of AMD, such as CEBPB and CFHR5, respectively (Fig. 2a, b), whereas proteins such as FUT5 were associated with all stages of AMD (Fig. 2a, b). We further stratified late AMD into GA or nAMD and compared protein quintiles of all AMD-associated proteins with the different AMD-related outcomes. Levels of some AMD-associated proteins, such as CFHR1, BPIFB1, and CFHR5, increased almost continuously from no AMD to advanced nAMD (Supplementary Fig. 2A–C). Supplementary Figure 3 depicts the distribution and relationship between the different quintiles of all 28 AMD-associated protein levels and AMD-related outcomes, while Supplementary Fig. 4 compares the top and bottom quintiles. Again, it is evident that while some proteins were associated with all AMD stages (e.g., CFHR1, NDUFS4), some were more, or only associated with early (e.g., LINGO1, RAB17) or late-stage AMD (e.g., BIRC2) (Supplementary Figs. 3 and 4). We note that the aptamer specificity for 15 of the AMD-associated proteins with sufficient pull-down mass spectrometry sensitivity was fully validated using orthogonal analysis (Methods section and Supplementary Data 4).

The serum protein network, closely linked to many common diseases and under genetic control, was recently identified[11]. This work showed that serum proteins exist in 27 coregulatory network modules, most of which are arranged in larger clusters. Many of the 28 AMD-associated proteins were correlated to each other (Supplementary Fig. 5), and in fact, 12 of these 28 proteins were enriched in a single module of the serum protein network ($P = 2.1 \times 10^{-8}$) (Fig. 2c and Supplementary Data 5). This module, known as serum protein module 13 (PM13), was not linked to any of the previously studied cardio-metabolic traits[11]. The Eigenprotein (the first principal component) of PM13 was found to be significantly associated with AMD-related outcomes (Fig. 2d). Serum protein modules included proteins synthesized in all solid tissues and may serve as an integration of the body's tissue-specific networks. Similarly, some AMD-related proteins were abundantly expressed in AMD-relevant ocular cells (Supplementary Fig. 6), whereas others were enriched in tissues such as the liver and brain (Supplementary Data 6).

**Proteins that indicate the progression of or predict late-stage AMD.** Using single point sex and age-adjusted logistic regression analysis, we examined which, if any, of the 4137 serum proteins in early AMD subjects only anticipated advancement to late AMD (pure GA or nAMD) in the same people over a 5-year follow-up period. Considering multiple comparisons, a single protein, PRMT3, showed significantly increased levels ($OR = 1.88$, $P = 5.3 \times 10^{-6}$) in early AMD at the baseline exam and prior to progression to pure GA at follow-up (Fig. 3a and Supplementary Fig. 7A). Figure 3b and Supplementary Fig. 7B show the observed five-year incidence of pure GA AMD across baseline PRMT3 quartiles, with the highest quartile having a significant increase in the number of new patients compared to the lowest ($P = 0.00047$), using an age and sex-adjusted logistic model. Following that, we looked at how the effect on progression varied across PRMT3 quartiles. As shown in Fig. 3c and Supplementary Fig. 7C, the effect appears to be additive from the lowest to the

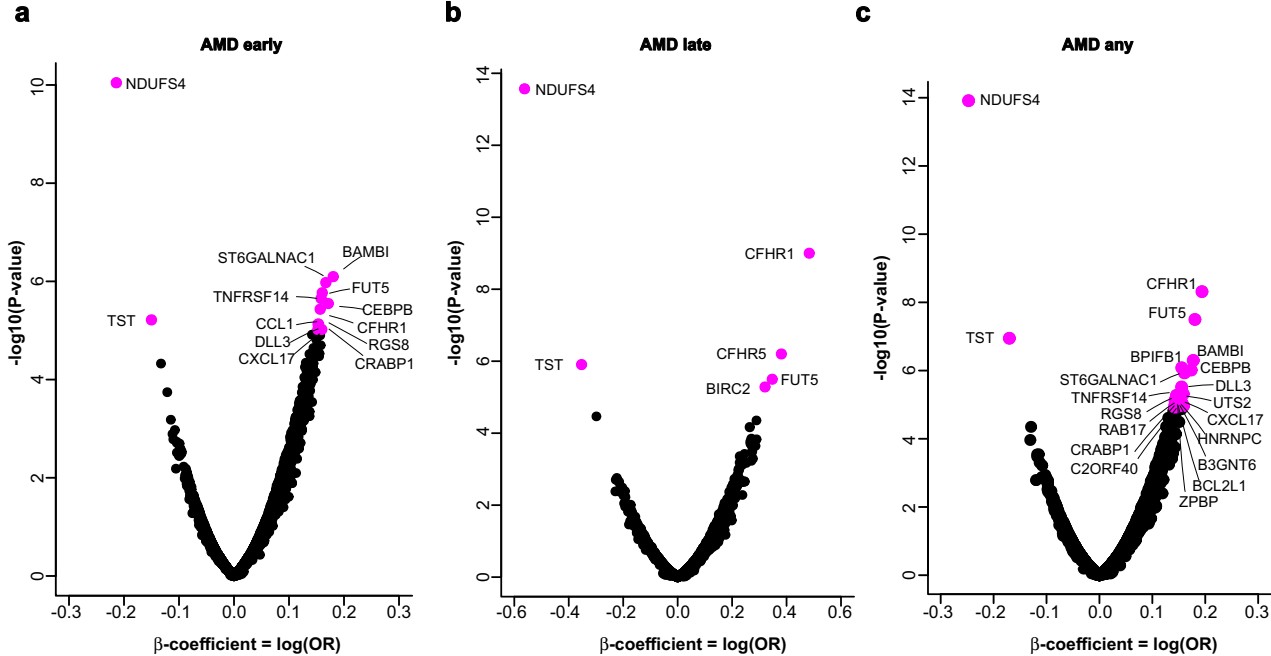

**Fig. 1 Association of global serum proteins with different stages of AMD. a** A volcano plot of all serum proteins associated with AMD early is shown using logistic regression analysis and Bonferroni correction for multiple comparisons, with colored (magenta) data points highlighting study-wide significant associations ($P < 1.21 \times 10^{-5}$, two-sided). Definition of early-stage AMD was according to Holliday et al.[22]. **b** Similar analysis and representative volcano plots of late-stage AMD, and **c** for AMD any (early and late stage AMD), where early-stage of the disease was defined according to Holliday et al.[22]. The y-axis represents the -$\log_{10}$(P-value) of associations (linear regression), while the x-axis represents the estimate as beta-coefficient = log(odds ratio).

highest levels of PRMT3 (F-test $P = 0.001$). No baseline protein changes predicted progression of early AMD to nAMD between first visit and five-year follow-up.

Age has been linked to the progression of both nAMD and GA AMD, whereas higher HDL cholesterol levels have been linked to the progression of GA and sex (females) with nAMD only[23]. Additionally, smoking has been linked to the rapid progression of GA AMD[24]. According to a recent study, known AMD susceptibility variants on chromosomes 1 and 10 can predict the progression of GA AMD[25]. The area under the curve (AUC) was estimated for pure GA prediction comparing models with and without PRMT3 as a predictor to determine if the protein improves prediction beyond known risk factors. Consequently, we examined all known risk factors for prediction of pure GA AMD development using estimates of AUC and included high impact genetic markers on chromosomes 1 (rs10922109 and rs570618) and 10 (rs3750846). This revealed that PRMT3 significantly improved the prediction of pure GA (AUC = 0.76, $P = 0.0249$) and was additive to all the parameters tested (Fig. 3d and Supplementary Fig. 7D).

Next, we applied a data-driven nonparametric bootstrap[26] and least absolute shrinkage and selection operator (LASSO)[27] to estimate the sampling distribution of logistic regression coefficients for all 4782 proteins in order to identify independent protein predictors of advanced AMD (Methods). The 21 proteins that appeared in at least 80% of the 500 iterations included the AMD-associated proteins CFHR1, CFHR5, FUT5, BIRC2, and NDUFS4 (Fig. 4). Here, CFHR1 and NDUFS4 had coefficients that were always non-zero in predicting late-stage AMD (Fig. 4). Interestingly, the complement factors CFH and CFB are among the protein predictors not listed in Table 1. Supplementary Figure 8 depicts a receiver operating characteristic curve (ROC) for the diagnostic ability of the 21 protein predictors to classify late-stage AMD, demonstrating a significant difference between the two ROC curves, that is the demographics versus the demographics plus proteins ROC curves (F-test of equality

$P = 1.4 \times 10^{-17}$). Overall, these findings highlight the additional benefit of using LASSO regression to uncover new aspects of the relationship between global serum proteins and AMD.

**Serum proteins link genetics to AMD, revealing its pathobiology.** In the most recent GWAS meta-analysis of advanced AMD, Fritsche et al.[10] examined 16,144 AMD patients, identifying 52 independent common and rare variants across 34 distinct genomic loci (Supplementary Data 7). We tested each of these variants for an effect on the 4782 proteins with the aim to narrow down the causal candidates at these genetic risk loci. Here, 18 of the risk loci for AMD were associated with a total of 340 serum proteins (Supplementary Data 7 and 8). In addition, the variant rs11080055, linked to the previously described[11] hotspot rs704 ($r^2 = 0.784$) was associated with several hundred serum proteins (Supplementary Data 7 and 9). Intriguingly, 22 out of 28 proteins found to be linked to AMD above were associated with one or more of six AMD susceptibility variants (Table 2). The AMD associated variant rs10922109 in the *CFH* gene at 1q31.3, for example, was significantly associated with AMD in the AGES-RS ($P = 2.6 \times 10^{-16}$) (Supplementary Data 7) and influenced the levels of 40 serum proteins (Fig. 5 and Supplementary Data 7 and 8), including the AMD-associated proteins CFHR1, TST, DLL3, ST6GALNAC1, CFP, and NDUFS4 (Fig. 5, Table 1, and Supplementary Data 8). Among the proteins associated with rs10922109, complement system proteins are significantly over-represented (Supplementary Data 10). These included the *cis* (proximal) associated proteins CFHR1, CFHR4, and CFH (Figs. 5 and 6a–c), as well as the *trans* (distal) associated proteins C3 (Fig. 5), CFP (Figs. 5 and 6d), and CFB (Fig. 5). According to the GTEx database[28], rs10922109 is associated with mRNA levels of the genes encoding CFHR1, CFHR4, and CFH (Supplementary Fig. 9A), with directionally consistent effects on the transcripts and cognate proteins (Supplementary Fig. 9A, B). CFHR4 and CFH were not significantly associated with AMD

**Table 1 Serum proteins significantly associated with prevalent AMD using logistic regression analysis.**

| Protein | AMD any | | | AMD early[a] | | | AMD late | | |
|---|---|---|---|---|---|---|---|---|---|
| | β-value | P-value | P adj.[b] | β-value | P-value | P adj.[b] | β-value | P-value | P adj.[b] |
| NDUFS4 | −0.298 | 1.4E-17 | 7.3E-14 | −0.260 | 3.0E-12 | 1.4E-08 | −0.562 | 2.7E-14 | 1.4E-10 |
| CFHR1 | 0.194 | 4.8E-09 | 2.5E-05 | 0.157 | 3.7E-06 | 0.01617 | 0.483 | 1.0E-09 | 5.0E-06 |
| FUT5 | 0.180 | 1.2E-08 | 0.00016 | 0.160 | 1.7E-06 | 0.00747 | 0.352 | 7.5E-07 | 0.00329 |
| TST | −0.235 | 2.6E-11 | 1.3E-07 | −0.218 | 8.1E-09 | 3.5E-05 | −0.353 | 1.2E-06 | 0.00544 |
| BAMBI | 0.177 | 5.0E-07 | 0.00255 | 0.158 | 8.2E-07 | 0.00357 | 0.138 | 0.06560 | ns |
| BPIFB1 | 0.193 | 3.1E-08 | 0.00016 | 0.176 | 1.9E-06 | 0.00733 | 0.271 | 0.00022 | ns |
| CEBPB | 0.173 | 9.7E-07 | 0.00489 | 0.171 | 2.6E-06 | 0.01239 | 0.189 | 0.01393 | ns |
| ST6GALNAC1 | 0.161 | 1.1E-06 | 0.00591 | 0.167 | 1.1E-06 | 0.00465 | 0.102 | 0.13511 | ns |
| DLL3 | 0.190 | 1.8E-07 | 0.00093 | 0.188 | 1.4E-06 | 0.00536 | 0.170 | 0.01847 | ns |
| UTS2 | 0.158 | 4.8E-06 | 0.02449 | 0.157 | 2.3E-05 | ns | 0.157 | 0.03319 | ns |
| TNFRSF14 | 0.147 | 5.3E-06 | 0.02696 | 0.158 | 2.2E-06 | 0.00977 | 0.123 | 0.09463 | ns |
| CXCL17 | 0.146 | 5.3E-06 | 0.02700 | 0.145 | 1.18E-05 | 0.04882 | 0.138 | 0.03687 | ns |
| RGS8 | 0.149 | 5.4E-06 | 0.02762 | 0.153 | 7.4E-06 | 0.03254 | 0.137 | 0.04946 | ns |
| RAB17 | 0.152 | 7.1E-06 | 0.03613 | 0.147 | 2.5E-05 | ns | 0.186 | 0.01027 | ns |
| HNRNPC | 0.143 | 7.4E-06 | 0.03769 | 0.137 | 3.0E-05 | ns | 0.163 | 0.01434 | ns |
| CRABP1 | 0.156 | 8.7E-06 | 0.04415 | 0.159 | 9.5E-06 | 0.04178 | 0.148 | 0.05272 | ns |
| CFHR5 | 0.161 | 1.1E-05 | 0.0463 | 0.077 | 0.01843 | ns | 0.406 | 3.3E-08 | 0.00014 |
| RPS6KB1 | 0.169 | 2.3E-06 | 0.00538 | 0.103 | 0.00093 | ns | 0.155 | 0.02414 | ns |
| ZPBP | 0.159 | 1.1E-05 | 0.04729 | 0.155 | 2.6E-05 | ns | 0.171 | 0.02920 | ns |
| KREMEN2 | 0.183 | 3.2E-06 | 0.01627 | 0.188 | 7.3E-06 | 0.02791 | 0.168 | 0.03047 | ns |
| CCL1 | 0.142 | 2.2E-05 | ns | 0.154 | 7.4E-06 | 0.03254 | 0.177 | 0.01066 | ns |
| CFP | −0.158 | 1.1E-05 | 0.04689 | −0.103 | 0.00079 | ns | −0.108 | 0.13210 | ns |
| GHR | −0.156 | 1.1E-05 | 0.04343 | −0.072 | 0.02341 | ns | −0.145 | 0.02935 | ns |
| LINGO1 | 0.162 | 8.0E-06 | 0.04634 | 0.120 | 0.00051 | ns | 0.172 | 0.01456 | ns |
| B3GNT6 | 0.142 | 1.0E-05 | 0.04602 | 0.104 | 7.2E-04 | ns | 0.201 | 0.00291 | ns |
| C2orf40 | 0.142 | 1.2E-05 | 0.04882 | 0.118 | 1.5E-04 | ns | 0.153 | 0.02070 | ns |
| BCL2L1 | 0.147 | 1.2E-05 | 0.04782 | 0.127 | 4.9E-05 | ns | 0.137 | 0.06061 | ns |
| BIRC2 | 0.107 | 0.00057 | ns | 0.086 | 0.00693 | ns | 0.321 | 5.0E-06 | 0.02279 |

*ns* not significant.

[a]Early stage AMD was defined by either Holliday et al.[22] or Jonasson et al.[23]. Based on either definition of early AMD, only the most significant association is reported. For more information, see Supplementary Data 2 and 3.

[b]The term P adj. refers to the P-value adjusted for the number of comparisons. All reported P-values are two-sided and derived from sex and age-adjusted logistic regression analysis.

using stringent multiple testing corrections, though the risk allele effect was directionally consistent with their relationships seen using the top and bottom quintiles and various stages of AMD (Fig. 6b, c). C3 and CFB, the two *trans* regulated proteins, were not significantly linked to any AMD-related outcomes (data not shown). Finally, out of the 40 proteins associated with rs10922109, 11 proteins cluster in protein module PM13 (Supplementary Data 11 and 12).

Aside from the expected enrichment of the complement system among the 340 proteins associated with AMD-linked variants (Supplemental data 8 and 10), there were many previously unknown links, including SLC5A8 (aka SMCT1). SLC5A8 is a $Na^+$-coupled transporter located in the retina and notable for its ability to transport the prodrug 2-oxothiazolidine-4-carboxylate (OTC) across the retina[29,30]. This protein was found to be linked to two AMD-associated variants, rs2070895 and rs2043085 in or proximal to the *LIPC* gene (Supplementary Data 8 and Fig. 6e), as well as the *APOB* intron variant rs2678379 (Fig. 6e), which is a well-established regulator of plasma lipoprotein (HDL, LDL, and triglycerides) levels[31]. The genetic influence and disease association show that serum levels of SLC5A8 are inversely related to AMD (Fig. 6f), which is consistent with the protein's proposed protective function[30]. The proteins linked to AMD-related genetic variants map to pathways with both known and previous unknown association to the disease (Supplementary Note 1 and Supplementary Data 10).

We examined the relationship between all known AMD-related GWAS variants and the 27 serum protein network modules through their Eigenproteins (1st and 2nd principal components).

The modules in supercluster III (modules PM11 to PM15)[11], were associated with the largest number of AMD-causing genetic variants (Supplementary Data 13). For example, the *CFH* variant rs570618 at 1q31.3, which is a proxy for the CFH Y402H (aka rs1061170) missense mutation, was linked to AMD risk (Supplementary Data 7) and is associated with 217 serum proteins, 100 of which are found in PM13 (Supplementary Data 8 and 11). Also, rs570618 was significantly associated with the Eigenprotein for the AMD-associated protein module PM13 (Supplementary Data 13), further reinforcing its connection to AMD. In passing, rs570618 was associated with 33 of the 40 proteins linked to rs10922109 (see above) and are both intronic SNPs in the *CFH* gene in a moderate linkage disequilibrium ($r^2 = 0.41$), but the latter is also associated with 184 other serum proteins (Supplementary Data 8).

The earliest recognized and most well-studied genetic factors for AMD were variants on chromosome 1 (1q31.3) spanning the *CFH* gene[5–8]. Further investigation into the AMD-related genetic effects of 1q31.3 variants, which included haplotype analysis, found a common deletion across the *CFHR3* and *CFHR1* genes that protects against AMD[32]. The eight variants across the *CFH* gene[10] are not completely independent of the *CFHR3-CFHR1* deletion[33], which is tagged by the variant rs6677604 (allele A)[34]. We looked at the effect of rs6677604 on global serum proteins as a proxy for the *CFHR3-CFHR1* deletion. The protective allele A for rs6677604 was significantly associated with reduced risk of late AMD in the AGES-RS cohort ($\beta = -0.04$, $P = 3.0 \times 10^{-7}$), and was also strongly linked with lower serum levels of CFHR1 ($\beta = -1.33$, $P = 2.7 \times 10^{-890}$; Supplementary Data 14). Overall,

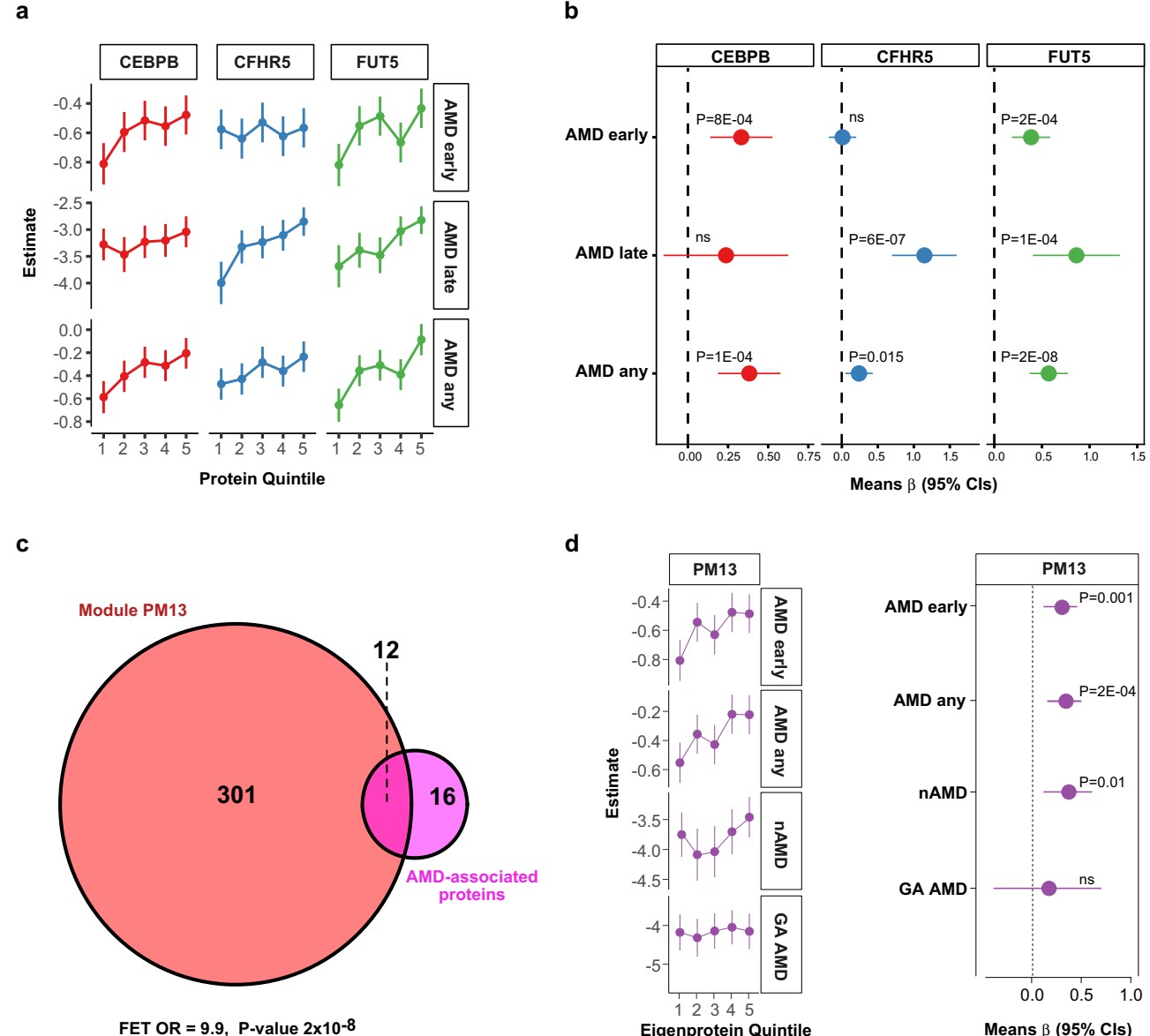

**Fig. 2 Serum proteins are linked to different stages of AMD. a** Relationship between quintiles of serum CEBPB, CFHR5, and FUT5 levels and AMD outcomes, whereas **b** highlights the comparison of the top and bottom quintiles of the same proteins to different AMD outcomes. **c** Venn diagram demonstrating significant enrichment for 12 of the 28 AMD-associated proteins (OR = 9.89, $P = 2.1 \times 10^{-8}$, two-sided) in the previously described serum protein network module called PM13[11]. FET is an abbreviation for Fisher exact test. **d** The Eigenprotein for PM13 was found to be associated with AMD-related outcomes such as nAMD but not GA AMD. Definition of early-stage AMD in **a**, **b**, **d** was according to Holliday et al.[22] and the number of patients in each AMD-related group is shown in Supplementary Data 1. The dependent variable is the AMD outcome, while the protein quintiles are predictor variables, and the logistic regression analysis was adjusted for age and sex. The data points in **a**, **d** (left panel) represent the predicted mean log odds of the AMD event with the quintiles treated as categorical variables, and the error bars represent 95% confidence intervals (CIs). The data points in **b**, **d** (right panel) are the mean estimate of beta in the logistic regression (logOR) using quintiles as continuous predictor variable and the error bars represent 95% CIs. All highlighted $P$-values are two-sided. NS denotes not significant ($P > 0.05$).

rs6677604 was linked to 22 proteins in serum (Supplementary Data 14), including six proteins that were not associated with any of the other AMD genetic markers listed in Supplementary Data 8 and 9. For instance, calnexin (CANX) was previously implicated in ARMS2 secretion[35] and was one of the six novel proteins associated with rs6677604 (Supplementary Data 14).

**Mendelian randomization analysis identifies proteins causally related to AMD.** To determine if any of the 28 AMD-associated proteins might be causally related to the disease, a two-sample

Mendelian randomization (MR) study was performed using *cis*-acting genetic variants as instruments. BPIFB1, CFHR1, CFHR5, FUT5, and GHR were found to have such instruments[13]. However, the genetic instruments for BPIFB1 and GHR do not overlap with previously known risk loci for AMD (Table 2). The causal estimate for CFHR1, CFHR5, and FUT5 was determined using the generalized weighted least squares method (GWLS)[36], where CFHR1, CFHR5, and FUT5 were found to be significant (FDR < 0.05; Fig. 7a). For both definitions of AMD (Methods), the causal estimate agrees with the observed relationship between these proteins and AMD (Fig. 7a), highlighting their role as risk

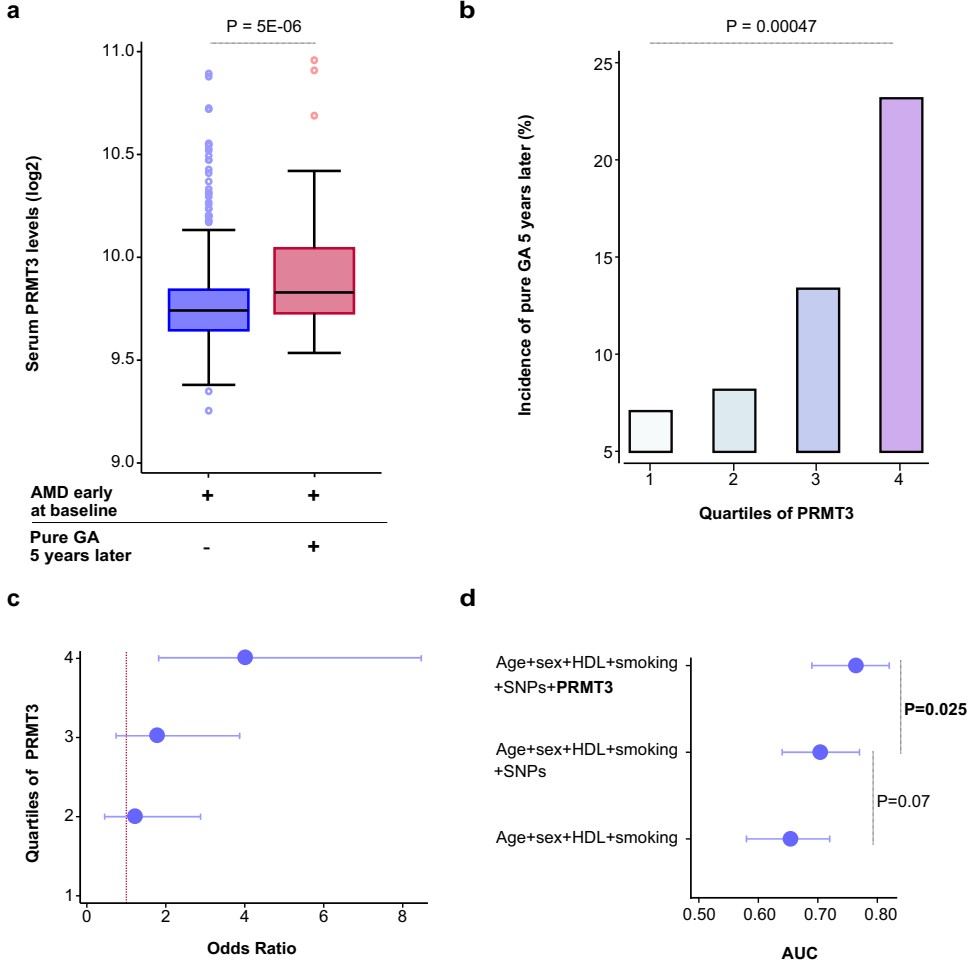

**Fig. 3 PRMT3 is a novel biomarker for the progression of geographic atrophy. a** Boxplot depicting the distribution of PRMT3 by progression status on a log2 scale. The blue box represents individuals ($n = 428$) who had early-stage AMD (definition by Jonasson et al.[23]) at baseline and after a five-year follow-up, while the red box represents subjects ($n = 62$) who progressed from early-stage AMD to pure GA. The boxplot indicates median value, 25th and 75th percentiles. Whiskers extend to smallest/largest value no further than 1.5× interquartile range with outliers being shown. The highlighted *P*-value is two-sided, obtained from age and sex-adjusted logistic regression analysis. **b** The observed percentage incidence of pure GA at five-year follow-up (*y*-axis), as represented by PRMT3 quartiles (*x*-axis), was assessed using an age and sex-adjusted logistic model. The *P*-value shown is two-sided. **c** Age and sex-adjusted odds ratios (OR) from a logistic regression model for progression to pure GA after a five-year follow-up by PRMT3 quartiles (1st quartile as reference) (F-test $P = 0.001$, two-sided). The error bars for the mean OR (center) are the 95% CIs. **d** Area under the curve (AUC) comparison for various logistic models with incident pure GA as the outcome variable and various predictors such as age, HDL cholesterol (HDL), smoking (current), AMD genetic risk variants (single-nucleotide polymorphisms; SNPs) at chromosomes 1 (rs10922109 and rs570618) and 10 (rs3750846), and PRMT3. The error bars for the mean AUC (center) are the 95% CIs. The highlighted *P*-values are two-sided.

factors for AMD development. Of the three proteins, CFHR1 was detected with ELISA in a much smaller sample of an independent study of healthy volunteers and AMD patients, confirming elevated circulating levels of CFHR1 in the disease's most advanced stage (Methods section and Supplementary Fig. 10). Significant increases in all factor H-related proteins, including CFHR1, have also been linked to AMD in recent studies[33,37]. Figure 7b–d shows scatter plots with the generalized weighted causal estimate and MR-Egger regression for CFHR1, CFHR5, and FUT5, which were all identified as causal candidates in the MR analysis. The *P*-values for the Egger intercept were >0.05, indicating that there was not statistically significant evidence of pleiotropy (Fig. 7b–d). Because the loci containing the genes encoding CFHR1, CFHR5, and FUT5 are saturated with multiple independent variants for both AMD (Supplementary Data 7) and proteins[13,14], colocalization analysis becomes difficult and thus inconclusive[38,39]. As a result, colocalization analyses were omitted from the current study.

In a secondary analysis, we repeated the MR analysis, but this time we included 1327 aptamers with *cis*-acting genetic instruments. In this study, eight additional proteins, including C3, CFI, AIF1, and VTN, were found to have a significant causal estimate for AMD (Supplementary Fig. 11). Here, the allograft inflammatory factor 1 (AIF1) was found to be significantly associated with AMD outcomes in this population and directionally consistent with the causal estimate after multiple testing correction (Supplementary Data 15). These proteins were collectively enriched for the complement and coagulation cascade (FDR = 0.0002). CFHR1 remained statistically significant at this more stringent multiple testing correction threshold (Supplementary Fig. 11).

## Discussion

Homeostasis has long been recognized as a property of living systems and is maintained by integrating local and global signals

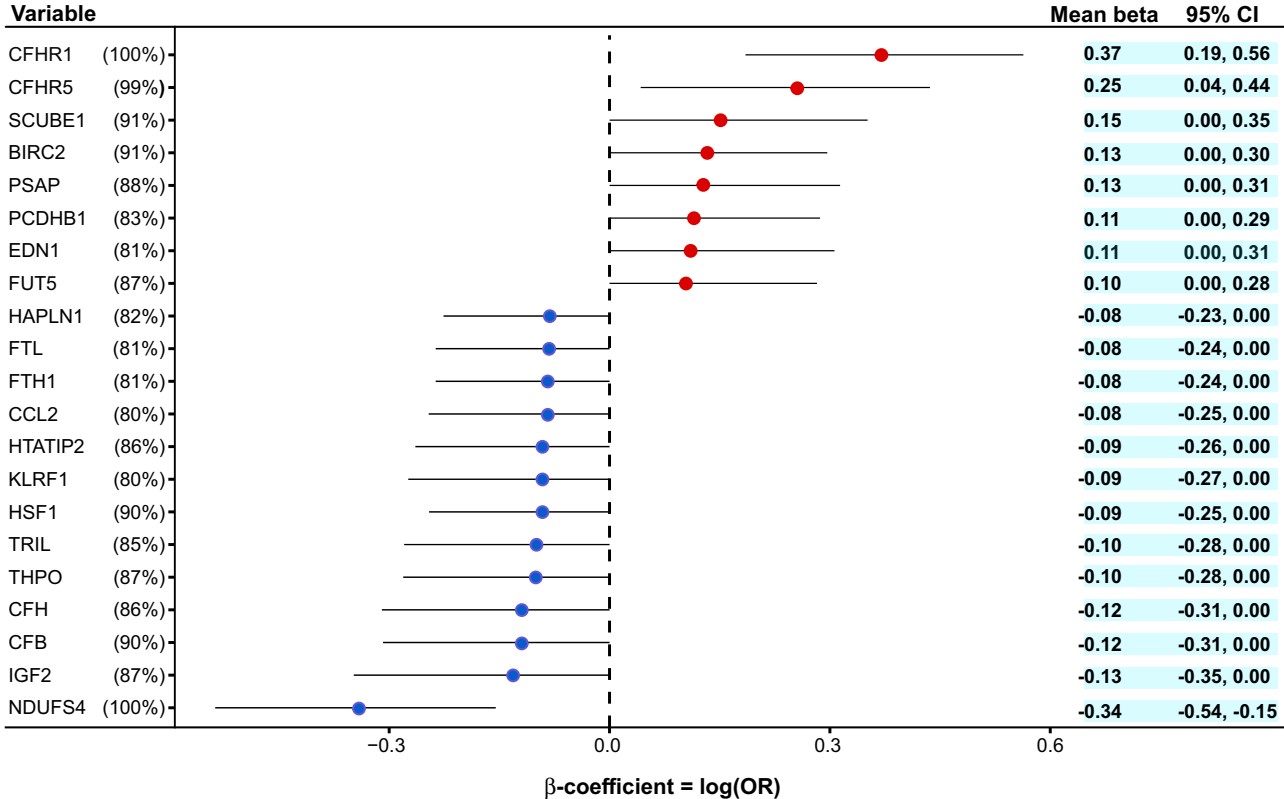

**Fig. 4 Identification of serum proteins that predict late-stage AMD.** The figure depicts protein predictors for late-stage AMD using a data-driven nonparametric bootstrap and LASSO regression analysis (see Methods section for more detail). On the left, we show 21 protein variables that appear in at least 80% of bootstrap iterations, while the mean estimates and 95% confidence intervals (CIs) are shown on the right. The percentage of iterations with non-zero coefficients for the protein variables is shown in parenthesis. Each center data point represents the mean beta-coefficient = log(odds ratio) with the 95% CIs as error bars.

so that each tissue does not act independently, but rather influences and responds to preserve the systemic equilibrium and coordination. The measurement of the levels of thousands of proteins in serum of thousands of individuals has begun to illuminate the process by which homeostasis may occur[11,12,19]. In this study we document how serum proteins report on and influence AMD appearance and progression in the eye. Previous studies of AMD patient plasma and urine have revealed changes in lipid and energy metabolites; however, sample size has limited detection of serological protein changes, or stratification of late AMD into GA and nAMD[40–45]. The population-based AGES-RS study with its array of biomarkers, clinical profiles, and genetic risk factors collected prospectively from participants who were aged over 67 at baseline visit, and a follow-up visit after five-years has enabled us to identify circulating proteins and protein networks in patients that are associated with AMD stage and progression. Several grading systems have been used to classify the disease stage of AMD. In our study, we used the classification described by Holliday et al.[22] as well as that of Jonasson et al.[23]. The distinction between these two systems is that one uses a stricter criterion for diagnosis of early AMD (Methods section), resulting in fewer cases. An additional less stringent system[22] was included to explore protein associations more fully. It is hoped that using the different criterion will facilitate cross-referencing to various studies in the public domain.

Consistent with the known links between AMD and the complement pathway, three complement pathway proteins were associated with AMD (CFHR1, CFHR5, and CFP). Six additional proteins known to modulate both the innate and adaptive

immune response were also associated with AMD (TNFRSF14, CCL1, CXCL17, BPIFB1, BIRC2, and CEBPB), implicating induction of inflammatory processes in AMD. Reductions in two mitochondrial proteins, the complex 1 protein NDUFS4 and the sulfotransferase protein TST were also associated with AMD, consistent with structural, functional, and genetic mitochondrial changes in AMD[46–50]. Inactivation of the NDUFS4 gene causes a severe form of the vision loss disease Leigh syndrome, and mice lacking this gene die prematurely at the age of 50-60 days[51], with compromised photoreceptor function[52] and excessive lipid droplet formation[53]. Elevation of the ribosomal S6 protein kinase 1 (RPS6KB1), a component of the nutrient-responsive mTOR (mammalian target of rapamycin) signaling pathway was also associated with AMD, consistent with the AMD-like phenotype in mice with RPE mTOR overactivation[54–56]. The pro-longevity effects of *RPS6KB1* gene deletion[57], or mTORC1 inhibition across multiple species[58], are also intriguing when considering AMD as a disorder of aging. Some of these proteins, such as CFHR1, CFHR5, BIRC2, and NDUFS4, were found to be among the 21 independent predictors of late-stage AMD.

The rate of progression from early to late AMD in the AGES-RS was ~4.5% per year[23]. Serum levels of PRMT3 protein were elevated in early AMD patients who subsequently progressed to GA, but not in those progressing to nAMD. PRMT3 controls ribosomal activity via arginine methylation of the 40S ribosomal subunit protein RPS2[59–61]. Interestingly, control of mRNA translation occurs via phosphorylation of another component of the 40S ribosomal subunit, RPS6, by the previously mentioned kinase RPS6KB1. Also, a third ribosomal 40S subunit RPS10 was

**Table 2 AMD-linked proteins associated with AMD genetic risk variants using linear regression.**

| Protein | AMD variant[a] | | Effect of variant on serum protein levels[b] | | |
|---|---|---|---|---|---|
| | SNP | Most proximal gene | $\beta$-value | P-value | pQTL |
| NDUFS4 | rs10922109 | CFH | 0.5854 | 9.86E-223 | Trans |
| | rs570618 | CFH | −0.8904 | 1.20E-566 | Trans |
| CFHR1 | rs10922109 | CFH | −0.5848 | 1.16E-234 | Cis |
| | rs570618 | CFH | 0.4675 | 1.37E-139 | Cis |
| FUT5 | rs12019136 | FUT6 | −0.9998 | 6.51E-86 | Cis |
| TST | rs10922109 | CFH | 0.5241 | 3.69E-178 | Trans |
| | rs570618 | CFH | −0.7866 | 1.36E-424 | Trans |
| BAMBI | rs570618 | CFH | 0.1364 | 1.03E-14 | Trans |
| BPIFB1 | None | N/A | N/A | N/A | N/A |
| CEBPB | rs570618 | CFH | 0.0959 | 4.59E-08 | Trans |
| ST6GALNAC1 | rs10922109 | CFH | −0.1018 | 3.12E-08 | Trans |
| | rs570618 | CFH | 0.1488 | 2.00E-15 | Trans |
| | rs11080055 | TMEM97 | −0.2254 | 7.21E-37 | Trans |
| DLL3 | rs10922109 | CFH | −0.1219 | 1.98E-11 | Trans |
| | rs570618 | CFH | 0.1593 | 7.24E-18 | Trans |
| UTS2 | rs570618 | CFH | 0.1301 | 3.88E-13 | Trans |
| | rs11080055 | TMEM97 | −0.1874 | 3.84E-28 | Trans |
| TNFRSF14 | rs570618 | CFH | 0.1272 | 2.14E-11 | Trans |
| | rs2230199 | C3 | 0.1693 | 3.96E-17 | Trans |
| CXCL17 | None | N/A | N/A | N/A | N/A |
| RGS8 | rs570618 | CFH | 0.1711 | 7.43E-20 | Trans |
| RAB17 | rs570618 | CFH | 0.1631 | 6.24E-19 | Trans |
| HNRNPC | rs11080055 | TMEM97 | −0.0957 | 2.18E-07 | Trans |
| CRABP1 | rs570618 | CFH | 0.1003 | 1.74E-08 | Trans |
| | rs11080055 | TMEM97 | −0.1854 | 5.16E-28 | Trans |
| CFHR5 | rs570618 | CFH | 0.2284 | 4.53E-32 | Cis |
| RPS6KB1 | None | N/A | N/A | N/A | N/A |
| ZPBP | rs570618 | CFH | 0.1016 | 3.81E-09 | Trans |
| | rs11080055 | TMEM97 | −0.1301 | 2.27E-15 | Trans |
| KREMEN2 | rs570618 | CFH | 0.0866 | 3.55E-07 | Trans |
| CCL1 | rs570618 | CFH | 0.1820 | 1.71E-22 | Trans |
| CFP | rs10922109 | CFH | 0.3049 | 2.64E-62 | Trans |
| | rs570618 | CFH | −0.3028 | 3.82E-59 | Trans |
| GHR | None | N/A | N/A | N/A | N/A |
| LINGO1 | None | N/A | N/A | N/A | N/A |
| B3GNT6 | None | N/A | N/A | N/A | N/A |
| C2orf40 | rs570618 | CFH | 0.1704 | 4.25E-19 | Trans |
| | rs11080055 | TMEM97 | −0.1099 | 1.68E-09 | Trans |
| BCL2L1 | rs570618 | CFH | 0.1611 | 3.94E-18 | Trans |
| BIRC2 | rs429358 | APOE | −0.5793 | 1.54E-106 | Trans |

*Cis* proximal effect, *Trans* distal effect, *N/A* not applicable.
[a]AMD associated variants are from Fritsche et al.[10] and highlighted in Supplementary Data 7.
[b]Reported *P*-values are two-sided derived from sex and age-adjusted linear regression analysis. The effect alleles for each corresponding pQTL[13] are shown in Supplementary Data 8 and 9. To account for multiple comparisons, significant associations with *P*-values of $<1 \times 10^{-6}$ are reported.

controlled by the AMD-associated variants rs570618 and rs429358. Alterations in post-translational regulation of protein synthesis by both phosphorylation and arginine methylation could be a key early driver of AMD pathogenesis and progression.

AMD is associated with 34 distinct genomic loci, accounting for nearly half of AMD's genetic variance[10]. Integrating intermediate traits such as mRNA and/or protein levels with genetics and disease traits helps in identifying the causal candidates[11,62–65]. The present study integrated the most recent findings of AMD genetic risk factors with 4137 human serum proteins, providing mechanistic insights into previously described AMD SNPs. Many of the variants that explained the largest fraction of the genetic susceptibility for AMD, were associated with the levels of numerous circulating proteins. Interestingly, of the AMD-associated proteins identified, the majority were linked to one or more of only six AMD susceptibility variants. The *CFH* variant rs10922109 (allele A) for example, confers AMD protection (OR = 0.51) and is associated with lower CFHR1 and CFHR4

protein levels and higher CFH levels. Consistent with these changes in protein levels, rs10922109 (allele C) has previously been associated with increased CFHR1[33], activation of the complement cascade in AMD patients[66] and also increased serum CFHR4[67]. AMD variants had both overlapping and distinct effects on serum proteins and were associated with numerous proteins in the complement cascade. For instance, the *CFH* variant rs570618 influenced 33 of the 40 proteins associated with the co-localized variant rs10922109 (e.g., CFHR1/4, C3, CFB, and CFP), but rs570618 is also associated with 184 other serum proteins (e.g., CFHR5, C1S and C8) (Supplementary Data 8). As previously noted, rs570618 is a surrogate for the CFH missense mutation Y402H. It has been claimed that this mutation causes CFH to bind less tightly to CRP[68], impairing debris clearance and increasing retinal inflammation. According to the current study, Y402H in CFH is associated with variations in blood levels of 217 proteins, which suggests a more complex explanation of the substantial risk of rs570618 for AMD. A previous study has connected the

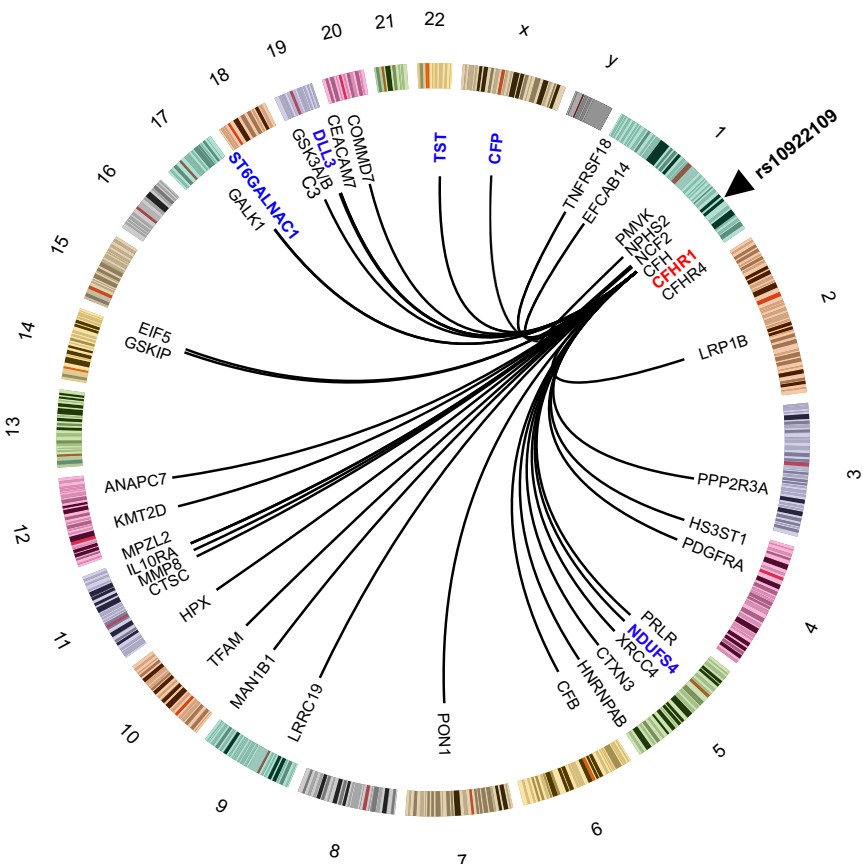

**Fig. 5 Serum proteins link genetics to AMD.** The Circos plot depicts the relationship between the AMD-associated variant rs10922109 at 1q31.3 and 40 serum proteins (see Supplementary Data 7 and 8 for details). The blue (*trans* effects) and red (*cis* effects) colored proteins are among the 28 AMD-associated proteins listed in Table 1.

rs570618 AMD risk allele T to elevated CFHR1 and CFHR5 levels[33], which is consistent with our findings (Supplementary Data 8). Finally, we examined the association of protein networks with AMD, since deep profiling has revealed the modular structure of the serum proteome containing 27 distinct protein modules[11]. Interestingly, the protein module PM13 is directly associated with AMD outcomes and is enriched for individual proteins linked to AMD (12 of 28) and AMD risk variants.

CFHR1 and CFHR5 are presumed pathogenic since they impede CFH binding to pro-inflammatory lipid peroxidation products[69,70], and induce inflammasome activation[71], whereas CFHR1 gene deletion is known to be protective for AMD[69]. Indeed, the two-sample MR test analysis revealed that both proteins were causally linked to AMD, consistent with prior MR analyses of factor H-related proteins found causally related to AMD[37]. However, because their *cis* regions overlap (Supplementary Data 8), from which the genetic instruments are selected, it is impossible to say whether one or both are the causal effector. In addition, FUT5 was supported as causal candidate in the MR analysis. As 23 of the AMD-associated proteins did not have any *cis*-acting instruments, they could not be tested in the MR analyses, and we thus cannot exclude their candidacy as causal proteins. In the extended MR analysis, eight additional proteins were supported as causal, including other members of the complement system (CFI, C3, VTN), further highlighting the role of this pathway in the development of AMD. In contrast to these strong genetic associations driving systemic protein changes discussed above, rs3750846 in the *ARMS2/HTRA1* locus on chromosome 10 was not associated with serum protein levels in the present study. One reason could be that the current platform lacks aptamers that detect the ARMS2 and HTRA1 proteins encoded by genes near rs3750846, but the variant is known to regulate the transcription of both *ARMS2* and *HTRA1* in solid tissues[72]. Alternatively, this variant may alter ocular specific protein changes that are not apparent in serum. It should be noted, however, that CANX, which was previously implicated in ARMS2 secretion[35], was linked to the AMD-associated variant rs6677604 on chromosome 1 (Supplementary Data 14).

AMD is a multifactorial age-related disease with a complex pathobiology in which systemic and local inflammatory and other effectors play significant roles. Recent evidence indicates that thousands of proteins present in serum participate in cross tissue regulation that connects all parts of the body[11,12]. This occurs by tissues secreting protein(s) into the bloodstream which control biological processes in other physically distant tissues, resulting in a network of cross-tissue regulatory loops[12]. For example, the expression of the proteins CFHR1 and CFHR5, which have been causally linked to AMD in this study, are highly specific to the liver, whereas FUT5 is expressed in the bone marrow and testis (Supplementary Data 6). More research is required to determine the roles of the numerous serum proteins discussed in this study and their potential effect on various pathophysiological processes that lead to different stages of AMD in the local eye setting. Some of the serum proteins associated with AMD in this study are characterized as intracellular proteins, and the significance of their presence in serum, remains to be determined. Our findings provide a comprehensive and unique framework for understanding the pathobiology of AMD, which may lead to the discovery of systemic biomarkers and therapeutic targets for the disease.

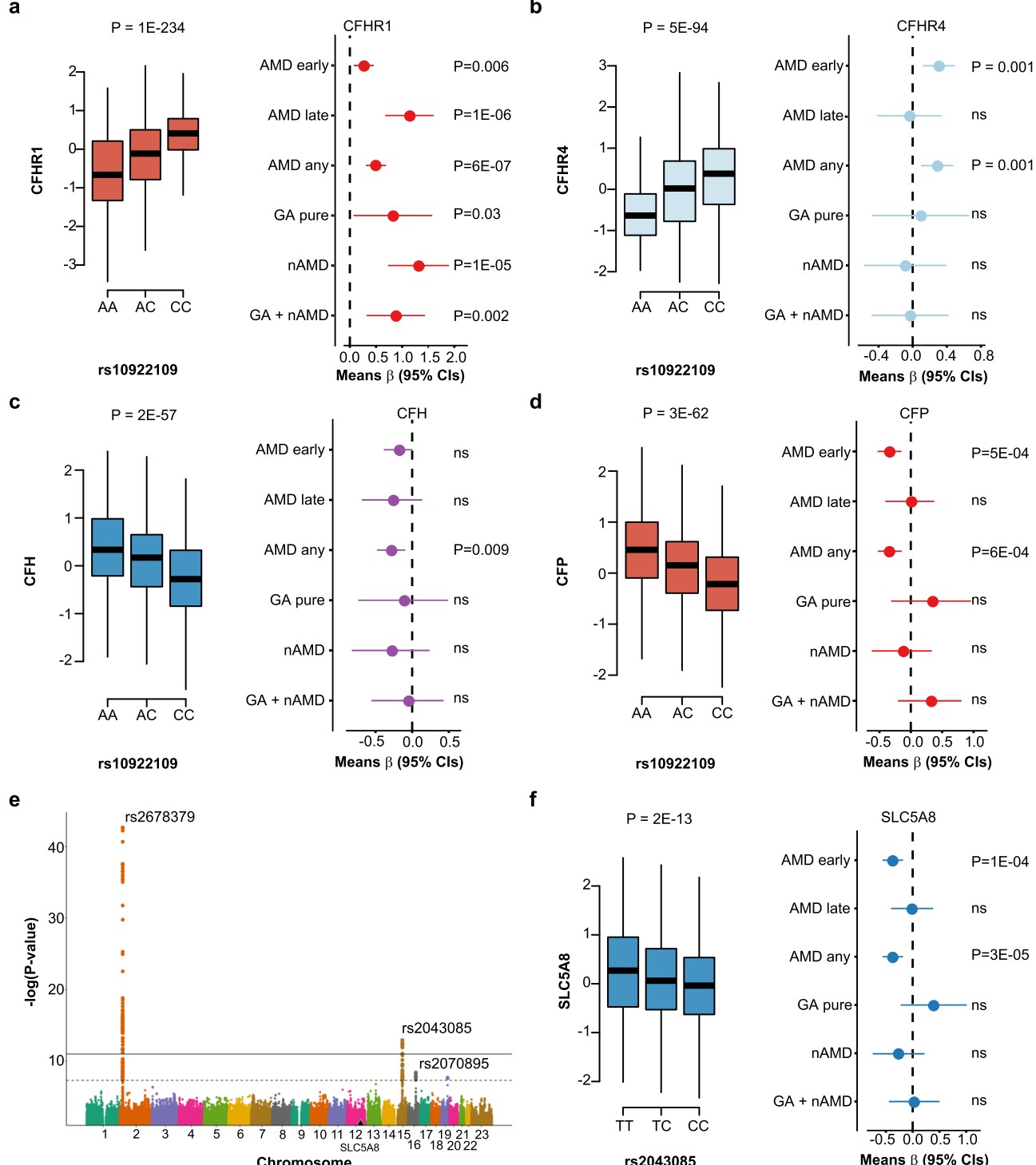

**Fig. 6 Serum proteins linked to AMD risk variants and AMD-related outcomes. a** Box plot showing serum levels of CFHR1 as a function of copy C alleles for the variant rs10922109 (left panel). The relationship between the 5th and 1st quintile (reference) of CFHR1 serum levels and various AMD related outcomes is shown in the right panel. **b–d** Similar plots for CFHR4, CFH and CFP. **e** The Manhattan plot highlights *trans*-acting variants at different chromosomes in a GWAS of serum SLC5A8 levels, specifically the AMD-associated variants rs2070895 and rs2043085, as well as the variant rs2678379, which affects lipoprotein levels. The *y*-axis shows the $-(\log_{10})$ of the *P*-values (two-sided) for the association of each genetic variant present along the *x*-axis at different chromosomes. **f** A box plot (left panel) of the AMD-associated variant rs2043085 affecting serum levels of SLC5A8 ($P = 2 \times 10^{-13}$, two-sided), and the relationship of the 5th and 1st (reference) quintiles of SLC5A8 levels with different AMD-related outcomes (right panel). All box plots in the figure show median (middle line), 25th, 75th percentile (box), and 5th and 95th percentile (whiskers). The relationship between SNPs and protein levels was studied using linear regression, while the protein relationship to AMD outcomes was examined using logistic regression. All regression analyses were age and sex adjusted. The data points in the right panels of **a–d**, **f** are the mean estimate of beta in the logistic regression (logOR) using quintiles as continuous predictor variable and the error bars represent 95% CIs. All highlighted *P*-values are two-sided. NS stands for not significant ($P > 0.05$). The number of patients in each AMD-related group is shown in Supplementary Data 1 and described in Methods section.

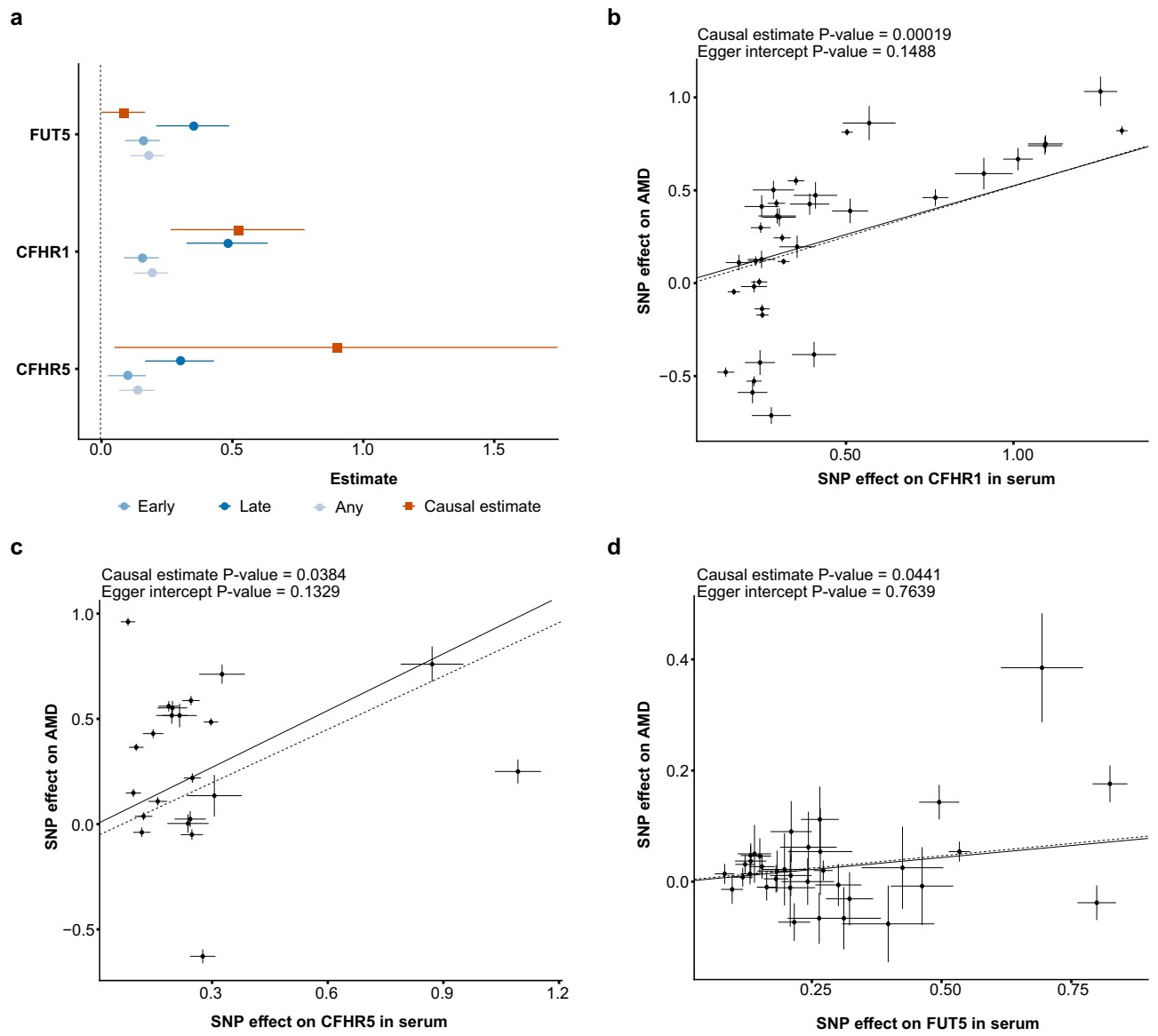

**Fig. 7 MR analysis identifies serum proteins that are causally related to AMD. a** The causal estimate (squares) from the two-sample MR analysis compared to the observational estimates (circles) for each of the five proteins with *cis*-acting instruments and associated with AMD in the observational study (AGES-RS, *n* = 5457). As each protein could have different observational coefficients depending on which definition of early AMD (see Methods section) was used, it was decided to select and display the coefficient for each definition which had the lower adjusted *P*-value. The causal estimator for CFHR1, CFHR5 and FUT5 was significant (FDR < 0.05) and positive. Each center data point shows the estimate as beta = log(OR) for the observational estimate and as described in Methods for the causal estimate, along with 95% confidence intervals as error bars. The number of patients in each AMD-related group in the AGES-RS cohort is shown in Supplementary Data 1. **b** Scatterplot for the CFHR1 protein supported as having a causal effect on AMD in a two-sample MR analysis. The figure demonstrates the estimated effects of the respective *cis*-acting genetic instruments on the serum CFHR1 levels in AGES-RS (*x*-axis) and risk of AMD through a GWAS provided by the IAMDGC consortium (*n* = 16,144 AMD patients)[10] (*y*-axis). The solid line indicates the generalized weighted causal estimate (*β* = 0.524, SE = 0.126, *P* = 0.00019, two-sided), while the dotted line shows the MR-Egger regression. Similar plots as in **b** are shown for **c** CFHR5 (*β* = 0.899, SE = 0.407, *P* = 0.0384, two-sided), and **d** FUT5 (*β* = 0.087, SE = 0.041, *P* = 0.0441, two-sided). Each data point in the center of the scatter plots in **b**–**d** represents the SNP effect (log(OR)) on disease with vertical lines as error bars (95% CIs) versus the SNP effect (beta-coefficient) on the protein with horizontal lines as error bars (95% CIs). The *P*-values (two-sided) for the Egger intercept and the GWLS causal estimates are displayed at the top of each scatter plot. The causal estimates for the three proteins in **b**–**d** were found to be significant after adjusting the *P*-value with the Benjamini–Hochberg method (see Methods section).

## Methods

**Study population**. Participants aged 66 through 96 were from the Age, Gene/Environment Susceptibility Reykjavik Study (AGES-RS) cohort[73]. The AGES-RS was approved by the NBC in Iceland (approval number VSN-00-063), and by the National Institute on Aging Intramural Institutional Review Board, and the Data Protection Authority in Iceland. AGES-RS is a single-center prospective population-based study of highly phenotyped subjects (5764, mean age 76.6 ± 5.6 years) and

survivors of the 40-year-long prospective Reykjavik study (N ~ 18,000), an epidemiologic study aimed to understand aging in the context of gene/environment interaction by focusing on four biologic systems: vascular, neurocognitive (including sensory), musculoskeletal, and body composition/metabolism. Descriptive statistics of this cohort as well as detailed definition of the various disease endpoints and relevant phenotypes measured have been published[11,73]. Of the 5764 AGES-RS participants 3411 attended a 5-year follow-up visit (AGES-RS II).

Detailed description of AMD diagnosis and the baseline characteristics of the AMD population in AGES-RS has previously been described in two separate publications[22,23], which vary in terms of the medical definition of early stage AMD. In Holliday et al.[22], early stage AMD ($n = 1755$) was defined as the presence of soft drusen (>63 μm) alone, retinal pigment epithelium (RPE) depigmentation alone or a combination of soft drusen with increased retinal pigment and/or depigmentation in the absence of late AMD, while Jonasson et al.[23] defined early AMD ($n = 1054$) by the presence of any soft drusen and pigmentary abnormalities (increased or decreased retinal pigment) or the presence of large soft drusen ≥125 μm in diameter with a large drusen area >500 μm in diameter, or large ≥125 μm indistinct soft drusen in the absence of signs of late AMD[23]. Late AMD ($n = 272$) was defined by the presence of any of the following: geographic atrophy (GA) or exudative AMD including subretinal hemorrhage, subretinal fibrous scar, RPE detachment, or serous detachment of the sensory retina or signs of treatment for neovascular AMD[23]. Also, early and late AMD was combined as AMD any. Late AMD was separated into GA pure ($n = 112$) or neovascular AMD (nAMD, $n = 160$). GA with possibly some exudative AMD was also defined (GA + nAMD, $n = 183$). We used both medical definitions of early AMD (and AMD any) separately to analyze the association of serum proteins with AMD.

**Protein measurements via SOMAmers.** For the AGES-RS, we used a distinct version of the SomaScan platform (Novartis V3-5K), based on the slow-off rate modified aptamer (SOMAmer) protein profiling technology[11,12,74]. The aptamers are small single-stranded 40-mer DNA oligomers with modified nucleic acids selected to specifically recognize target proteins in their native three-dimensional state and show slow dissociation kinetics ($t_{1/2} > 30$ min) which in combination with stringent wash steps impedes nonspecific binding[74]. The custom-design SOMAs-can platform was built to quantify 5034 protein analytes in a single serum sample with a focus on proteins that are known or expected to be present extracellularly or on the surface of cells, 4782 of which SOMAmers directly bind to 4137 different human proteins. Blood samples were collected at the AGES-RS baseline, after an overnight fast. Serum was prepared using a standardized protocol[75], stored in 0.5 ml aliquots at −80 °C and serum samples that had not been previously thawed were used for the protein measurements. To avoid batch or time of processing biases, the order of sample collection and processing for protein measurements were randomized and all samples run as a single set at SomaLogic Inc. (Boulder, CO, US). All SOMAmers that passed quality control had median intra-assay and inter-assay coefficient of variation (CV) < 5% or equivalent to reported variability[76]. Several metrics, including aptamer specificity through direct tandem mass spectrometry (MS) analysis and inferential assessment via genetic analysis, have been used to determine the performance of the proteomic platform, suggesting strong target specificity throughout the platform[11]. Hybridization controls were used to correct for systematic variability in detection and calibrator samples of three dilution sets (40%, 1%, and 0.005%) were included so that the degree of fluorescence was a quantitative reflection of protein concentration. Box-Cox transformation was applied on the protein data[77], and extreme outlier values excluded, defined as values above the 99.5th percentile of the distribution of 99th percentile cutoffs across all proteins after scaling, resulting in the removal of an average 11 samples per SOMAmer. Previous studies have shown that protein quantitative trait loci (QTLs) replicate well across different study populations as well as proteomic platforms[11,21]. While a recent comparison of protein measurements across different platforms showed a wide range of correlations[78], *cis* pQTLs detection and validation by orthogonal MS-based measures were predictive of a strong correlation across platforms and were great indicators of platform specificity when protein concentrations obtained by orthogonal methods differ. The aptamer specificity of six of the AMD-associated proteins listed in the main text has already been validated by orthogonal mass spectrometry (MS)-based approach[11]. Nine additional aptamers were confirmed with SOMAmer pull down and MS (SP-MS) using AMD patient serum samples (Supplementary Data 4). Aside from low abundance affecting pull-down, there are many reasons why some aptamer-enriched proteins are not detected by SP-MS. For instance, MS analyses have different sensitivity than that of SOMAmer scan and is dependent on the protein sequence. Factors such as ion suppression of individual peptides caused by instrument/interfering molecules, incomplete trypsin digestion, and modifications of peptides either naturally or artifactually can limit the detection.

**Protein measurements via ELISA.** Participants aged 55 and over (up to $n = 60$, 15 per group), enrolled in a prospective study by Ophthalmic Consultants of Boston, to measure complement and disease-related protein biomarkers in blood. AMD patients with Age-Related Eye Disease Study (AREDS)[79] grade 2–4 had best corrected visual acuity (BCVA) of at least 20/200. Healthy volunteers were aged matched (±2 to AMD patients) with BCVA of 20/40 or better in both eyes, and a comprehensive eye examination within previous 12 months revealing no diagnosis other than refractive error, mild cataract, dry eye, or AREDS grade 1. AREDS grading criteria for grade 2 included mild changes including multiple small drusen, non-extensive intermediate drusen, and/or pigment abnormalities. AREDS grade 3 included at least 1 large drusen of at least 125 μm in diameter, extensive intermediate drusen, and/or noncentral geographic atrophy. AREDS grade 4 included advanced AMD central geographic atrophy in one or both eyes. Qualifying subjects completed an informed consent form, and no identifying patient information was included. A blood sample of 10 ml was collected by routine phlebotomy, in EDTA-coated, lavender-topped collection tubes. Plasma was isolated after centrifugation at 3184 × *g* for 30 min, frozen and stored at −80 °C. CFH antibody was purchased from Quidel (cat. #A255). Rabbits were immunized with recombinant human CFHR1 protein at Covance (Princeton, NJ), and CFHR1 cross-reactive antibodies were purified from serum using CFHR1-conjugated CNBr-activated Sepharose 4B resin (Cytiva, Marlborough, MA). Additionally, antibodies showing cross-reactivity to Factor H were removed by several rounds of depletion using Factor H-conjugated resin. CFHR1 levels in patient plasma were measured in a Meso Scale Discovery assay (Rockville, MD) using the selective antibodies.

We investigated any possible cross-reactivity of CFHR1 antibodies with the CFH and CFHR proteins. For this, recombinant CFH protein was obtained from Complement Technology (TX, US), and recombinant CFH and CFHR proteins were expressed in HEK cells. More specifically, anti-CFH antibodies demonstrated no cross-reactivity to CFHR proteins as measured by ELISA, whereas anti-CFHR1 antibodies showed trace cross-reactivity to CFHR2, however, with CFHR2 binding signal below the signal level of CFHR1 used to extrapolate unknown plasma levels.

**Analysis of gene expression in single-cell RNA sequencing data from eye tissues.** Two separate single-cell RNA-sequencing experiments on seven human donor eyes, five controls and two AMD patients, yielded the data for single-cell gene expression[80], also found at the GEO database (https://www.ncbi.nlm.nih.gov/geo/query/acc.cgi?acc=GSE135922). The first study assessed single cells from the RPE/choroid (2 controls and 1 AMD), whereas the second evaluated the endothelial population after a CD31 antibody enrichment step (3 controls and 1 AMD)[80]. Each of the AMD donors was labeled as having "neovascular AMD," with no other phenotypic information provided. The normalized single-cell data was downloaded from GEO and was analyzed with the R package Seurat (v.3.0.0) in R 3.6.3 environment. The final dataset contained 4335 cells after filtering. Variable genes were identified using Seurat with default parameters and Principal Components Analysis (PCA) was performed on these variable genes. First 11 PCs of the single-cell data (resolution = 0.2) were used for clustering cells with similar gene expression profile. Clusters were identified using FindNeighbors and FindClusters functions from Seurat package and UMAP dimensionality reduction was utilized for cluster visualization. The cell clusters were then manually annotated based on the markers reported in the paper[80].

**Statistical and genetic analysis.** We used linear or logistic regression for the associations of individual proteins as well as Eigenvectors of protein modules with various phenotype and genotype measures, depending on the result being continuous or binary. Statistical results were obtained using linear models for continuous outcomes and generalized logistic models for binary outcomes for comparison of protein quintiles and AMD-related clinical traits. The models were fit using the outcome phenotypes as dependent variables and protein quintiles as predictor variables along with the adjustment variables age and sex. The protein quintiles were treated as factor variables to avoid underlying assumptions regarding linearity of effects. Continuous outcomes were standardized using z-scores prior to model fitting. Thus, coefficient estimates should be interpreted on the standard deviation scale. That is an estimated mean difference of 1 between protein quintiles translates to a one-standard-deviation difference between groups after adjusting for other included variables. The expected means were obtained as linear predictions from the fitted models along with the fitted confidence intervals around the mean. The linear predictions for qualitative phenotypes are shown on the log-odds scale. The difference between the 5th and 1st protein quintiles was obtained as the expected marginal difference between those groups. Thus, for continuous outcomes, they are the optimal linear estimator with corresponding confidence intervals and *P*-values, but for discrete outcomes they are obtained using the commonly applied asymptotic approximations. We applied linear regression using an additive genetic model for all single-point SNP association analyses of different disease-related outcomes as well as for proteins. The results of associating genetic variants with serum protein levels were obtained using a GWAS of 4782 human proteins measured in serum using 7,506,463 assayed and imputed genetic variants from 5368 AGES-RS individuals[13].

For identification of protein predictors for late-stage AMD, we approximated the sampling distribution of logistic regression coefficients for all 4782 protein variables through the nonparametric bootstrap[26] and least absolute shrinkage and selection operator (LASSO)[27], estimated using the glmnet package for R[81]. We summarized our results after 500 bootstrap iterations by calculating the coefficients mean, 95% confidence intervals by calculating the 2.5% and 97.5% quantiles, and the percentage of iterations in which they were non-zero. We adjusted the LASSO models for age and sex in our analysis by leaving their coefficients unpenalized. Here, we fit the logistic regression LASSO model for prevalent late-stage AMD and compare the odds of late AMD to no AMD. All statistical analyses were conducted using the software environment R statistical package version 3.6.0 (2019-04-26) and and RStudio (1.1.456).

**Two-sample Mendelian randomization analysis.** In a two-sample Mendelian randomization study, genetic variants for each trait, the protein-encoding gene (exposure) *X* and AMD (outcome) *Y* are found in two distinct samples. A genetic

variant (SNP) $Z$ is used in the analysis provided it fulfills the three assumptions of instrumental variables:

1. There exists a significant association between SNP $Z$ and exposure $X$.
2. SNP $Z$ is independent of any confounder $U$ which might influence exposure $X$ and outcome $Y$.
3. SNP $Z$ is independent of outcome $\underline{Y}$ conditional on exposure $X$ and confounder $U$. This assumption is usually referred to as the exclusion criterion.

The first assumption is readily tested by setting a threshold on the significance level on the SNP-exposure association. Unfortunately, it remains a challenge to examine the validity of the second and third assumptions. When a SNP violates the third assumption, we generally speak of pleiotropy. To obtain the SNP-exposure associations, all genetic instruments within a 1 Mb (±500 kb) *cis* window for the protein-encoding gene were obtained for a given SOMAmer. A cis-window-wide significance level $P_b = 0.05/N$, where $N$ was the number of SNPs within a given *cis*-window, was computed. Genetic instruments within the *cis* window for each SOMAmer were then clumped such that variants in high linkage disequilibrium (LD) ($r^2 \geq 0.2$) within a 1 Mb region were combined. The list of variants was then further pruned by removing all instruments with $P \geq P_b$.

Summary statistics on the genetic risk on AMD were obtained from a GWAS provided by the IAMDGC consortium[10]. Any SNP in the *cis* window-wide significant data set not found in the AMD GWAS data set were replaced by proxy SNPs ($r^2 > 0.8$) when possible, to maximize SNP coverage. Causal estimate for each protein was obtained by the generalized weighted least squares (GWLS) method[36], which accounts for the correlation that can exist between instruments. Let $Z = \{1, 2, ..., M\}$ be an index set of all SNPs associated with a protein-encoding gene and AMD. For $j \in Z$, denote the $j$-th SNP-protein association and SNP-AMD association as $\beta_{X_j}$ and $\beta_{Y_j}$ respectively, each with their corresponding standard error $\sigma_{X_j}$ and $\sigma_{Y_j}$. Then, the causal estimate $\hat{\theta}$ is found by evaluating:

$$\hat{\theta} = \left(\beta_X^T \Sigma^{-1} \beta_X\right)^{-1} \beta_X^T \Sigma^{-1} \beta_Y, \quad (1)$$

where $\beta_X = \left(\beta_{X_1}, ..., \beta_{X_M}\right)^T$ and $\beta_Y = \left(\beta_{Y_1}, ..., \beta_{Y_M}\right)^T$ and $\Sigma^{-1}$ is the inverse of the weighting matrix $\Sigma$ whose $(i,j)$-th entry is $\rho_{ij}\sigma_{Y_i}\sigma_{Y_j}$ and $\rho_{ij}$ is the correlation between SNPs $i, j \in Z$. The standard error of $\hat{\theta}$ is $se(\hat{\theta}) = \alpha\sqrt{\left(\beta_X^T \Sigma^{-1} \beta_X\right)^{-1}}$, where $\alpha = \max\left(1, \sqrt{MSE}\right)$ with:

$$MSE = \frac{\left(\beta_Y - \beta_X\hat{\theta}\right)^2}{M - 1}, \quad (2)$$

reflects our uncertainty about the weights by assuming that we only know their relative magnitudes.

Finally, each estimate was subjected to a two-step sensitivity analysis. Any protein with a causal estimate found to be significant after adjusting the $P$-value with the Benjamini–Hochberg method was reassessed with the weighted median estimator which allows for up to 50% of the genetic instruments to violate any of the three assumptions of instrumental variables[82]. If the direction of the weighted median estimator was consistent with the GWLS estimate and remained significant at $P < 0.05$ it was subjected to the second stage of the sensitivity analysis but otherwise removed. Any protein which passed the first step was then reexamined with MR-Egger estimator which replaces the exclusion criterion with a weaker condition, the InSIDE assumption[83], which allows SNPs to exhibit pleiotropic effects provided that the pleiotropic effect is uncorrelated with the SNP-AMD effect $\beta_{Y_j}$. As with the first step, any protein whose MR-Egger estimate was directionally consistent with the GWLS estimate and had a $P$-value exceeding 0.05 for the intercept coefficient was kept and considered a causal candidate. Causality for proteins with single *cis*-acting variants was assessed with the Wald ratio estimator $\hat{\theta} = \beta_Y/\beta_X$.

**Reporting summary**. Further information on research design is available in the Nature Research Reporting Summary linked to this article.

## Data availability
The custom-design Novartis SOMAscan data are available through a collaboration agreement with the Novartis Institutes for BioMedical Research (lori.jennings@novartis.com). Data from the AGES Reykjavik study are available through collaboration (AGES_data_request@hjarta.is) under a data usage agreement with the IHA. All access to data is controlled *via* the use of a subject-signed informed consent authorization. The time it takes to respond to requests varies depending on their nature and circumstances of the request, but it will not exceed 14 working days. All data supporting the conclusions of the paper are presented in the main text and freely available as a supplement to this manuscript (Supplementary Information and Supplementary Data 1–15). The GTEx database (https://gtexportal.org/home/) was used to obtain the eQTL data.

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

## Acknowledgements

The authors acknowledge the contribution of the Icelandic Heart Association (IHA) staff to the AGES-RS, as well as the involvement of all study participants. We thank the IAMDGC consortium for supplying us with their GWAS summary statistics data. National Institute on Aging (NIA) contracts N01-AG-12100 and HHSN271201200022C for V.G. financed the AGES study; retinal image collection and AMD readings were

funded by the NIH Intramural Research Program (ZIAEY000401). V.G. received a funding from the NIA (1R01AG065596), and IHA received a support from Althingi (the Icelandic Parliament). The Icelandic Research Fund (IRF) funded V.E. and Va.G. with grants 195761-051, 184845-053, and 206692-051, while Va.G. received a postdoctoral research grant from the University of Iceland Research Fund

## Author contributions

V.E. and T.E.W. designed and supervised the study. V.E., T.E.W., Va.G., T.J., E.F.G., B.G.J., T.A., M.T., N.F., S.P., X.L., R.E., Y.Z., S.J., C.L.H., S.M.L., J.L., C.L.G., A.A.N., B.L., R.P., T.E.W., Vi.G., M.F.C., Z.L., L.J.L., L.L.J., F.J., Q.Z., and Q.H. performed data analysis and/or contributed to the acquisition of data. N.F., R.P., X.L., J.R.L., and L.L.J. provided expertize on proteomics data. V.E. and T.E.W. wrote the first draft of the manuscript, with all coauthors contributing to data interpretation and manuscript editing. All coauthors have approved the submitted version of the paper.

## Competing interests

The study was supported by the Novartis Institute for Biomedical Research. M.T., N.F., S.P., X.L., R.E., Y.Z., S.J., C.L.H., S.M.L., J.L., C.L.G., A.A.N., B.L., R.P., Z.L., L.L.J., T.E.W., Q.Z., Q.H., and J.R.L. are employees and stockholders of Novartis. All other authors have no conflict of interests to declare.
