## [Peer Review File · Nature Communications]

A Proteogenomic Signature of Age-related Macular Degeneration in BloodREVIEWER COMMENTS

Reviewer #1 (Remarks to the Author):

This study described a proteomics approach towards age-related macular degeneration, and linking the identified proteins with genetic variants associated with AMD. Overall, the study is interesting and the results have merit.

I have several remarks to improve the quality of the manuscript.

Results:

Line 95: Please explain why two distinct definitions of early-stage AMD were chosen. Many of the associations were identified in early AMD using one definition but not both (Table 1), which would question which definition should be followed. How should these results be interpreted for the different disease definitions? These different outcomes for the different classifications should at least be mentioned in the results and/or discussion sections.

Line 114, figure 1g: PRMT3 showed significantly increased levels in early AMD at baseline prior to progression of GA at follow-up. Were these results found for both classification systems of early AMD? It would be helpful so see a survival analysis: how does the progression rate in individuals with the highest PRMT3 quartile (or quintile) relate to those in the lowest quartile? Is the effect additive for the four quartiles? If yes, this could further confirm PRMT3 as a biomarker for progression. Several prediction models have been published for AMD progression, based on clinical, demographic and genetic markers. How predictive is PRMT3 for AMD progression (what is the AUC), and how does this relate to the AUC of prediction models using clinical, demographic and/or genetic markers?

Line 144: the statement that the 22 proteins are controlled by one or more of the six AMD susceptibility variants is too strongly stated. This study does not prove that the protein levels are controlled by these variants; they may be regulated by other regulatory /feedback processes at protein level. Controlled by should therefore be replaced by associated with.

Line 152: rs10922019 is not independent of the CFHR1-3 deletion (proxied by rs6677604). Both are located on haplotypes H3 and H7; while only haplotype H2 carries the alternative allele for rs10922019 but not the CFHR1-3 deletion. Performing a haplotype analysis would provide more insight on the effect of the CFHR1-3 deletion on CFHR1 and other complement protein levels, rather than rs10922019.

Line 172: Here it would be helpful to state that rs570618 is a proxy for CFH Y402H. This would help the scientific community realize that the effect of Y402H extends much further than only a amino acid change in the CFH protein, as the variant is associated with altered levels of 217 serum proteins. Some discussion on this in the Discussion section would also be helpful. Is the amino acid change Y402H causative in AMD, or could other proteins be involved rather than (or in addition to) the amino acid change in CFH?

Please note my previous statement also applies here to line 171: the statement that the variant controls protein levels is too strong: the variant is associated with altered protein levels. The same applies to line 175: 'influenced' and 'affected' are too strong terms. This should be replaced by associated, and should be applied in the whole manuscript. Same applies to lines 178 and 180: regulated is too strongly used here. Please soften the tone in the entire manuscript; regulation cannot be claimed here, only association.

Lines 175-177: as mentioned above, a haplotype analysis at the CFH locus would be more informative here. rs570618 and rs10992109 are not entirely independent, while their effects can better be disentangled by a haplotype analysis.

Lines 249-250: different effects of the C3 aptamers could be well explained by them targeting either C3 or degradation products of C3. It would be helpful to understand which parts of the C3 protein and which of the C3 degradation products are recognized by the different C3 aptamers.

Discussion

Line 306: the rs10922109 variant is described as an AMD protective variant in AMD, with the minor A allele occurring more frequently in controls than in AMD cases, with a reported OR of 0.51 in Fritsche et al. Therefore, the text needs to be rephrased here, stating that the protective A allele is associated with decreased levels of CFHR1 and CFHR4.

Lines 311-317: as mentioned above, the discussion on rs570618 should be extended, as it is a proxy for CFH Y402H. The results presented here imply that the effect of rs570618 (and thus Y402H) is not merely an amino acid change in the CFH protein, but rather an effect on many different proteins in various biological processes. It is important to point this out, as many scientists still hold on to the rather simplistic view that the CFH Y402H amino acid change is the cause of AMD.

The results should also be placed into context with two recently published studies describing an analysis of the FHR proteins in AMD: Cipriani et al *Am J Hum Genet* 2021 Aug 5;108(8):1385-1400 and Lores-Motta et al *Am J Hum Genet* 2021 Aug 5;108(8):1367-1384.

Methods, line 438: Antibodies for CFHR1 were evaluated for cross-reactivity to FH. CFHR1 is also highly similar to CFHR2, 3, 4 and 5. Crossreactivity against these other CFHR proteins should also be tested in order to confirm that the antibody recognized CFHR1 only. An approach to exclude crossreactivity would be to stratify individuals for the CFHR1-3 deletion (proxied by rs6677604). Given the frequency of the CFHR1-3 deletion in the population, several homozygous carriers of the deletion should be present in the dataset, and should have absent CFHR1 protein levels.

Reviewer #2 (Remarks to the Author):

Emilsson and colleagues performed analysis to associate serum protein levels with different types of AMD in the Ages cohort, followed by mapping genetic variants that related to both serum protein levels and risk of AMD, and finally, used MR to identify AMD causal proteins. They highlighted three proteins, CFHR1, CFHR5, and FUT5 to be causally linked to AMD. This comprehensive study leverages large proteomics data and by mapping them with the genetic architecture of AMD, adds to our understandings of the disease pathobiology. I have a few comments regarding the paper and particularly its presentation, please find them below.

Major comments:

I found the two sections of pathway analysis (lines 136-232) too long and convoluted. From what I understand, the authors map the risk variants of AMD to pQTLs, and many of these AMD risk variants (particularly variants at CFH locus) are highly pleiotropic in serum proteins (ie. Only 6 SNPs accounted for 22 of the 28 protein-AMD associations). And the rationale of section "serum proteins regulated..." is not very clear, it seems like this section is more of a discussion of the previous one. Figures 2b-e and figure 3 also seem a bit repetitive (ie. the titles for Figure 2 and 3 look like they are showing the same thing), is it possible to merge these figures? Sentences describing the gene/protein functions should be moved to discussion to make the core message clear and easy to read.

Because the proteins in the same cluster are likely not independent to each other, it is expected that most closely linked proteins (like in PM13) tend to also associate with the same outcome. Methods such as lasso/elastic net could pick independent proteins, which can be an interesting addition to the paper, particularly if the highlighted proteins are selected by lasso.

12 of the 28 proteins are in PM13, which modules the rest of the proteins belong to? Is the

Eigenprotein for PM13 also associated with other AMD outcomes? What about Eigenproteins for other modules?

For MR, based on the information in table 2, it is easy to guess that the three proteins with cis-SNPs that are also AMD risk variants are the proteins influencing AMD risk (That's why we usually do MR before the association analysis). However, the pleiotropic nature of these SNPs should also be discussed, particularly, the CFH locus SNPs are cis to multiple proteins which are all associated to AMD. In addition, the authors should perform colocalization analysis to exclude possibilities that MR results are confounded by LD.

It should also be interesting to include the protein-AMD association results for 21 MR significant proteins. If only looking at 20+ proteins instead of 4000+, some of them might survive multiple correction.

Minor comments:

Please use OR and 95% CI for logistic regression results.

Table S12: I expect to see the association of 52 AMD risk variants with 4000+ proteins, why the table seems to only have 4 variants?

In line 198-199, which are the "previously identified pathways and new pathways"? It's better to include those information in a table (or supplementary table), because it's not very clear in the text that follows.

Line 160-161: Possible citing error for Figure 2b.

There are still some legends missing from some supplementary tables. Ie. What's AUC in Table S2-3?

Response to Reviewers

We are pleased to submit our revised manuscript entitled “*A Proteogenomic Signature of Age-related Macular Degeneration in Blood*” (NCOMMS-21-31042-T) for consideration to be published in *Nature Communications*. We are grateful for the reviewers’ thoughtful and insightful comments. As you will see in the accompanying document, we have carefully considered, and addressed each of the comments point-by-point. Due to the significant amount of new analysis and the request to move text from the results to the Discussion section, we had to rearrange the main text and include new figures, as well as create a new Supplementary Note 1, Supplementary Data and Supplementary Figures. Finally, during the manuscript revision process, we carefully reviewed the text, tables, and all figures in accordance with the Nature Communications formatting guidelines. We hope you will agree that the incorporation of these changes has resulted in a considerably stronger paper.

Please note that we have included a new coauthor Dr. Zhiguang Li who has contributed significantly to the additional analyses of the revised manuscript. We have added his affiliation to the revised manuscript.

Responses below are provided in blue font. Text added to the revised manuscript has been italicized. Page and paragraph numbers listed below refer to the position of the text in the *clean* version of the revised manuscript (submitted along with a manuscript text file highlighting all changes using the track changes mode).

REVIEWER COMMENTS

Reviewer #1 (Remarks to the Author):

This study described a proteomics approach towards age-related macular degeneration and linking the identified proteins with genetic variants associated with AMD. Overall, the study is interesting and the results have merit.

I have several remarks to improve the quality of the manuscript.

Results:

Line 95: Please explain why two distinct definitions of early-stage AMD were chosen. Many of the associations were identified in early AMD using one definition but not both (Table 1), which would question which definition should be followed. How should these results be interpreted for the different disease definitions? These different outcomes for the different classifications should at least be mentioned in the results and/or discussion sections.

Response: We completely agree with the reviewer that this should be highlighted more prominently in the main text and elsewhere. In the literature, several grading systems have been used to classify AMD disease stages. In our study, we used the classification described by Holliday et al.¹ as well as that of Jonasson et al.² In early-stage AMD one group (Jonasson et al.²) uses a stricter/narrower diagnosis (see Methods), which results in fewer cases. The rationale for this was that drusen and, to a lesser extent pigmentary changes, can be transitory, and therefore not always indicative of established AMD disease. We employed this more stringent criteria to capture a fewer number of early AMD cases with higher confidence that those early lesions would not regress and that they would proceed to advanced AMD over time. In contrast, the Holliday et al.¹ classification system, yielded larger number of individuals classified with early AMD. Protein association studies found that 5 proteins were associated with early AMD as defined by Jonasson et al.², and 13 proteins using Holliday et al.¹ classification, with 3 in common. The different classification of early AMD naturally resulted in different numbers with AMD any, resulting in 13 proteins associated with any AMD using Jonasson et al.² and 20 proteins using Holliday et al.¹. We previously reported the prevalence, five-year incidence, and risk factors associated with AMD in this Icelandic population using the Jonasson et al.² classification; thus, reporting these data is useful to facilitate cross-referencing to these different public domain studies. To explore AMD classification more completely, we felt it appropriate to also employ the Holliday et al.¹ classification system as well. To better explain these analyses, we made the following changes to the results section:

Results, page 4 to 5, lines 89-95:

“Using sex- and age-adjusted logistic regression analysis of 4782 proteins (4137 gene symbols) and two distinct definitions of early-stage AMD (see Methods for details), we discovered that 28 serum proteins were associated with different stages of AMD using a study-wide significance

threshold (Fig. 1a-c, Table 1, Supplementary Fig. 1A-C, and Supplementary Data 2 and 3). This included 15 proteins associated with early-stage AMD identified by both Holliday et al.¹ and Jonasson et al.², with two proteins unique to Jonasson et al.² and ten proteins unique to Holliday et al.¹ (Fig. 1a, Supplementary Fig. 1A)”

Discussion, page 12, lines 267-272:

“Several grading systems have been used to classify the disease stage of AMD. In our study, we used the classification described by Holliday et al.¹ as well as that of Jonasson et al.². The distinction between these two systems is that one uses a stricter criterion for diagnosis of early AMD (Methods), resulting in fewer cases. An additional less stringent system¹ was included to explore protein associations more fully. It is hoped that using the different criterion will facilitate cross-referencing to various studies in the public domain.”

Finally, as suggested, we highlight which definition was used whenever we utilize only one.

Line 114, figure 1g: PRMT3 showed significantly increased levels in early AMD at baseline prior to progression of GA at follow-up. Were these results found for both classification systems of early AMD? It would be helpful so see a survival analysis: how does the progression rate in individuals with the highest PRMT3 quartile (or quintile) relate to those in the lowest quartile? Is the effect additive for the four quartiles? If yes, this could further confirm PRMT3 as a biomarker for progression. Several prediction models have been published for AMD progression, based on clinical, demographic and genetic markers. How predictive is PRMT3 for AMD progression (what is the AUC), and how does this relate to the AUC of prediction models using clinical, demographic and/or genetic markers?

Response: We appreciate the reviewer's thoughtful comments, and we've added further analysis as requested. In our original edition, we presented the association of PRMT3 with progression to advanced GA, using Jonasson et al.² definition of early AMD, because they examined progression of AMD in their work. We regret not being more explicit about this. We now show that PRMT3 has a substantial association with progression to GA using both definitions of early AMD. The new findings with PRMT3 are shown in a new section called "*Proteins that indicate the progression of or predict late-stage AMD*" and including two new figures, one in the main text (Fig. 3a-d, using the Jonasson et al.² definition of early AMD) and one Supplementary Fig. 7A-D (based on the Holliday et al.¹ definition of early AMD). Figure 3a-d is also shown below. Consequently, the Result section has been updated with new information about PRMT3 as a new progression biomarker for GA AMD.

More specifically, we find that PRMT3 was associated with progression from early AMD to the GA form of advanced AMD with odds ratio (OR) = 1.88 (95% CI; 1.43 to 2.47, P = 5.3×10⁻⁶) using Jonsson et al.² definition (Fig. 3a), and with OR = 1.71 (95% CI; 1.33 to 2.19, P = 2.8×10⁻⁵) using the Holliday et al.¹ definition of early AMD (Supplementary Fig. 7A). Further, Figure 3b and Supplementary Fig. 7B show the observed five-year incidence of pure GA AMD across baseline PRMT3 quartiles, with the highest quartile having a significant increase in the number of new patients compared to the lowest (P = 0.00047), using an age and sex adjusted logistic model. Following that, we looked at how the effect (OR) on progression varied across PRMT3 quartiles. As shown in Fig. 3c and Supplementary Fig. 7C, the effect appears to be additive from the lowest to the highest levels of PRMT3 (F-test P = 0.001).

Age has been linked to the onset and progression of both nAMD and GA AMD, whereas higher HDL cholesterol levels have been linked to the progression to GA and sex (females) with nAMD only². Furthermore, smoking has been linked to the rapid progression of GA AMD³. According to a recent study, known AMD susceptibility variants on chromosomes 1 and 10 can predict the progression to GA AMD⁴. The area under the curve (AUC) was estimated for pure GA prediction with and without PRMT3 adjustment to determine if the protein improves prediction beyond known risk factors. As a result, we examined all risk factors for prediction of pure GA AMD development using estimates of AUC including high impact genetic markers on chromosomes 1 (rs10922109 and rs570618) and 10 (rs3750846). This revealed that PRMT3 significantly improved the prediction of pure GA (AUC = 0.76, P = 0.0249) and was additive to all the parameters tested (Fig. 3d, Supplementary Fig. 7D).

The following text has been added to the Result section on pages 6-7, lines 120-144, to highlight these new results.

“Using single point sex and age-adjusted logistic regression analysis, we examined which if any of the 4137 proteins anticipated advancement to late AMD (pure GA or nAMD) while still in early AMD (using both definitions) in the same people over a 5-year follow-up period. Considering multiple comparisons, a single protein, PRMT3, showed significantly increased levels (OR = 1.88, P = 5.3×10⁻⁶) in early AMD at the baseline exam and prior to progression to pure GA at follow-up (Fig. 3a, Supplementary Fig 7A). Figure 3b and Supplementary Fig. 7B show the observed five-year incidence of pure GA AMD across baseline PRMT3 quartiles, with the highest quartile having a significant increase in the number of new patients compared to the lowest (P = 0.00047), using an age and sex adjusted logistic model. Following that, we looked at how the effect on progression varied across PRMT3 quartiles. As shown in Fig. 3c and Supplementary Fig. 7C, the effect appears to be additive from the lowest to the highest levels of PRMT3 (F-test P = 0.001). No baseline protein changes predicted progression of early AMD to nAMD between first visit and five-year follow-up.

Age has been linked to the progression of both nAMD and GA AMD, whereas higher HDL cholesterol levels have been linked to the progression of GA and sex (females) with nAMD only². Additionally, smoking has been linked to the rapid progression of GA AMD³. According to a recent study, known AMD susceptibility variants on chromosomes 1 and 10 can predict the progression of GA AMD⁴. The area under the curve (AUC) was estimated for pure GA prediction comparing models with and without PRMT3 adjustment to determine if the protein improves prediction beyond known risk factors. Consequently, we examined all known risk factors for prediction of pure GA AMD development using estimates of AUC and included high impact genetic markers on chromosomes 1 (rs10922109 and rs570618) and 10 (rs3750846). This revealed that PRMT3 significantly improved the prediction of pure GA (AUC = 0.76, P = 0.0249) and was additive to all the parameters tested (Fig. 3d, Supplementary Fig. 7D)”

Fig. 3. PRMT3 is a new biomarker for the progression of pure geographical atrophy (GA/dry AMD). **a** Boxplot depicting the distribution of PRMT3 by progression status on a log₂ scale. The blue box represents individuals who had early-stage AMD (definition by Jonasson et al.²) at baseline and follow-up, while the red box represents people who progressed from early-stage AMD to pure GA. The boxplot indicates median value, 25th and 75th percentiles. Whiskers extend to smallest/largest value no further than 1.5 × interquartile range with outliers being shown. **b** Observed incidence of pure GA at five-year follow-up (y-axis), presented by quartiles of PRMT3 (x-axis). **c** Age and sex adjusted odds ratios (OR) for progressing to pure GA after a five-year follow-up by PRMT3 quartile (1st quartile as reference), P-value from an F-test = 0.001. **d** Area under the curve (AUC) comparison for various models with incident pure GA as the outcome variable and various predictors such as age, HDL cholesterol, smoking, AMD genetic risk variants (SNPs) at chromosomes 1 (rs10922109 and rs570618) and 10 (rs3750846), and PRMT3.

Line 144: the statement that the 22 proteins are controlled by one or more of the six AMD susceptibility variants is too strongly stated. This study does not prove that the protein levels are controlled by these variants; they may be regulated by other regulatory /feedback processes at protein level. Controlled by should therefore be replaced by associated with.

Response: We accept the reviewer's viewpoint and have used associated with instead of controlled by (or regulated by, affected by, and so on; see response to related comment below) throughout the manuscript.

Line 152: rs10922019 is not independent of the CFHR1-3 deletion (proxied by rs6677604). Both are located on haplotypes H3 and H7; while only haplotype H2 carries the alternative allele for rs10922019 but not the CFHR1-3 deletion. Performing a haplotype analysis would provide more insight on the effect of the CFHR1-3 deletion on CFHR1 and other complement protein levels, rather than rs10922019.

Response: The earliest recognized, most well-studied, and strongest genetic factors for AMD were genetic variants on chromosome 1 (1q31.3) spanning the *CFH* gene⁵⁻⁸. Further investigation into the AMD-related genetic effects of 1q31.3 variants, which included haplotype analysis, found a common deletion across the *CFHR3* and *CFHR1* genes that protects against AMD⁹. In the largest GWAS of advanced AMD to date¹⁰, eight separate AMD risk variants at the 1q31.3 locus were identified, and these, along with 44 additional independent variants across 33 genomic regions, were evaluated in the current investigation for an effect on 4782 proteins in circulation. These eight variants including rs10922019, are not totally independent of the CFHR3-CFHR1 deletion, as the reviewer properly pointed out and is also recently highlighted in Lore's-Motta et al.¹¹. Here

the AMD protective haplotypes H3 and H7¹¹, which are the only haplotypes connected to reduced CFHR1 levels, carry the CFHR3-CFHR1 deletion¹¹, which is also tagged by the variant rs6677604 (allele A).

Rather than conducting an in-depth H3 and H7 haplotype analysis at the 1q31.3 region as suggested by the reviewer, which we believe would be better served as a separate study given all the additional analyses now included in an already comprehensive study covering 34 chromosomal regions including 52 independent SNPs, we examined the effect of rs6677604, as a proxy for the CFHR3-CFHR1 deletion, on levels of global serum proteins. The protective allele A for rs6677604 was significantly linked to reduced risk of late AMD in the AGES-RS cohort ($\beta = -0.04$, $P = 3 \times 10^{-7}$), and was also strongly associated with lower serum levels of CFHR1 ($\beta = -1.33$, $P < 1 \times 10^{-300}$) (new Supplementary Data 13). Overall, rs6677604 was linked to 22 proteins in serum, including six proteins that had not been linked to any of the other AMD genetic markers listed in Supplementary Data 7 and 8. For instance, calnexin (CANX) was previously implicated in ARMS2 secretion¹² and was one of the six novel proteins associated with rs6677604 (Supplementary Data 13).

These findings have been highlighted in a new Supplementary Data 13 and discussed in the main text at page 10, lines 207-220

“The earliest recognized and most well-studied genetic factors for AMD were variants on chromosome 1 (1q31.3) spanning the CFH gene⁵⁻⁸. Further investigation into the AMD-related genetic effects of 1q31.3 variants, which included haplotype analysis, found a common deletion across the CFHR3 and CFHR1 genes that protects against AMD⁹. The eight variants across the CFH gene¹⁰ are not completely independent of the CFHR3-CFHR1 deletion¹¹, which is tagged by the variant rs6677604 (allele A)¹³. We looked at the effect of rs6677604 on global serum proteins as a proxy for the CFHR3-CFHR1 deletion. The protective allele A for rs6677604 was significantly associated with reduced risk of late AMD in the AGES-RS cohort ($\beta = -0.04$, $P = 3 \times 10^{-7}$), and was also strongly linked with lower serum levels of CFHR1 ($\beta = -1.33$, $P = 1 \times 10^{-554}$) (Supplementary Data 13). Overall, rs6677604 was linked to 22 proteins in serum, including six proteins that were not associated with any of the other AMD genetic markers listed in Supplementary Data 7 and 8. For instance, calnexin (CANX) was previously implicated in ARMS2 secretion¹² and was one of the six novel proteins associated with rs6677604 (Supplementary Data 13)”

In addition, on Discussion page 16, lines 344-346

“It should be noted, however, that CANX, which was previously implicated in ARMS2 secretion¹², was linked to the AMD-associated variant rs6677604 on chromosome 1 (Supplementary Data 13)”

Line 172: Here it would be helpful to state that rs570618 is a proxy for CFH Y402H. This would help the scientific community realize that the effect of Y402H extends much further than only a amino acid change in the CFH protein, as the variant is associated with altered levels of 217 serum proteins. Some discussion on this in the Discussion section would also be helpful. Is the amino acid change Y402H causative in AMD, or could other proteins be involved rather than (or in addition to) the amino acid change in CFH?

Response: This is a good point, which we have addressed in both the Results and Discussion sections as follows:

Results, page 9, lines 198 to 201:

“For example, the CFH variant rs570618 at 1q31.3, which is a proxy for the CFH Y402H (aka rs1061170) missense mutation, was linked to AMD risk (Supplementary Data 6) and is associated with 217 serum proteins, 100 of which are found in PM13 (Supplementary Data 7 and 10)”

Discussion, page 14, lines 316 to 320:

“As previously noted, rs570618 is a surrogate for the CFH missense mutation Y402H. It has been claimed that this mutation causes CFH to bind less tightly to CRP¹⁴, impairing debris clearance and increasing retinal inflammation. According to the current study, Y402H in CFH is associated with variations in blood levels of 217 proteins, which suggests a more complex explanation of the substantial risk of rs570618 for AMD”

Please note my previous statement also applies here to line 171: the statement that the variant controls protein levels is too strong: the variant is associated with altered protein levels. The same applies to line 175: 'influenced' and 'affected' are too strong terms. This should be replaced by associated, and should be applied in the whole manuscript. Same applies to lines 178 and 180: regulated is too strongly used here. Please soften the tone in the entire manuscript; regulation cannot be claimed here, only association.

Response: See our response above to a similar comment: we accept the reviewer’s viewpoint and have used associated with instead of controlled by (or regulated by, affected by, and so on; see response to related comment below) throughout the manuscript.

Lines 175-177: as mentioned above, a haplotype analysis at the CFH locus would be more informative here. rs570618 and rs10992109 are not entirely independent, while their effects can better be disentangled by a haplotype analysis.

Response: We refer to our previous response to a similar request.

Lines 249-250: different effects of the C3 aptamers could be well explained by them targeting either C3 or degradation products of C3. It would be helpful to understand which parts of the C3 protein and which of the C3 degradation products are recognized by the different C3 aptamers.

Response: We agree with the reviewer that the disparities in results could be due to these aptamers' crossover binding to different C3 breakdown products. In this case, the C3 aptamer 2683-1_2 produced a positive causal estimate in Supplementary Fig. 11, whereas 4480-59_2 produced a negative causal estimate. Indeed, whereas 2683-1_2 binds primarily C3 with traces of C3b/iC3b, 4480-59_2 was less specific for C3 since it also bound iC3b and C3b (see R Fig. 1 below). Neither aptamer bound C3c or C3d (R Fig. 1 below).

R Fig. 1 Specificity of the C3 aptamers ELISA. Binding of increasing concentrations of aptamers 2683-1_2 (left panel) and 4480-59_2 (right panel) to complement proteins coated on 96-well plates. These data confirm 2683-1_2 binds C3, but not C3c/d, with minimal binding to C3b/iC3b. 4480-59_2 bound C3 and iC3b, and C3b with lower affinity. Like 2683-1_2, 4480-59_2 did not bind C3c or C3d.

We have amended the text in the results at page 11 to 12, lines 249 to 252:

“Different aptamers for C3 and VTN produced opposite effects (Supplementary Fig. 11), potentially due to aptamers binding different protein epitopes and/or isoforms, or breakdown products in the case of C3”

Discussion

Line 306: the rs10922109 variant is described as an AMD protective variant in AMD, with the minor A allele occurring more frequently in controls than in AMD cases, with a reported OR of 0.51 in Fritsche et al. Therefore, the text needs to be rephrased here, stating that the protective A allele is associated with decreased levels of CFHR1 and CFHR4.

Response: The current lines 307-309, page 14, have been changed because of this suggestion:

“The CFH variant rs10922109 (allele A), for example, confers AMD protection (OR = 0.51) and is associated with lower CFHR1 and CFHR4 protein levels and higher CFH levels.”

Lines 311-317: as mentioned above, the discussion on rs570618 should be extended, as it is a proxy for CFH Y402H. The results presented here imply that the effect of rs570618 (and thus Y402H) is not merely an amino acid change in the CFH protein, but rather an effect on many different proteins in various biological processes. It is important to point this out, as many scientists still hold on to the rather simplistic view that the CFH Y402H amino acid change is the cause of AMD.

Response: See our previous response to a similar comment in the preceding section. Here, Discussion, page 14, lines 316 to 320, has been updated with the following text.

“As previously noted, rs570618 is a surrogate for the CFH missense mutation Y402H. It has been claimed that this mutation causes CFH to bind less tightly to CRP, impairing debris clearance and increasing retinal inflammation. According to the current study, Y402H in CFH is associated with

variations in blood levels of 217 proteins, which suggests a more complex explanation of the substantial risk of rs570618 for AMD”

The results should also be placed into context with two recently published studies describing an analysis of the FHR proteins in AMD: Cipriani et al Am J Hum Genet 2021 Aug 5;108(8):1385-1400 and Lores-Motta et al Am J Hum Genet 2021 Aug 5;108(8):1367-1384.

Response: We appreciate the reviewer bringing this to our attention. We've modified the manuscript to reflect our findings considering these two recent papers. The following changes have been amended:

The following modifications have been made,

to lines 233 – 234 at page 11:

“Significant increases in all factor H-related proteins, including CFHR1, have also been linked to AMD in recent studies^{11,15}.”

to lines 309 – 311 at page 14:

“Consistent with these changes in protein levels, rs10922109 (allele C) has previously been associated with increased CFHR1¹¹, activation of the complement cascade in AMD patients¹⁶ and also increased serum CFHR4¹⁷”

to lines 320 – 322 at page 14 to 15:

“A previous study has connected the rs570618 AMD risk allele T to elevated CFHR1 and CFHR5 levels¹¹, which is consistent with our findings (Supplementary Data 7).”

to lines 329 – 331 at page 15:

“Indeed, the two-sample MR test analysis revealed that both proteins could be causally linked to AMD, consistent with prior MR analyses of factor H-related proteins found causally related to AMD¹⁵”

Methods, line 438: Antibodies for CFHR1 were evaluated for cross-reactivity to FH. CFHR1 is also highly similar to CFHR2, 3, 4 and 5. Crossreactivity against these other CFHR proteins should also be tested in order to confirm that the antibody recognized CHFR1 only. An approach to exclude crossreactivity would be to stratify individuals for the CFHR1-3 deletion (proxied by rs6677604). Given the frequency of the CFHR1-3 deletion in the population, several homozygous carriers of the deletion should be present in the dataset and should have absent CFHR1 protein levels.

Response: We appreciated this suggestion, which is a validation on subjects with the CFHR1-3 deletion; however, because shipping samples to Novartis wasn't a viable option, instead we performed additional antibody validation studies. For this, recombinant CFH (FH) protein was obtained from Complement Technology (TX, US), and recombinant FH and CFHR (FHR) proteins were expressed in HEK cells. More specifically, anti-FH antibodies demonstrated no cross-reactivity to FHR proteins as measured by ELISA (left panel in R Fig. 2 below), whereas anti-FHR1 antibodies showed trace cross-reactivity to FHR2 (right panel R Fig. 2 below), however the

FHR2 binding signal was below the signal level of FHR1 used to extrapolated unknown plasma levels.

In the Method section, lines 449-454, page 20-21, we have added a sentence to emphasize this:

“We investigated any possible cross-reactivity of CFHR1 antibodies with the CFH and CFHR proteins. For this, recombinant CFH protein was obtained from Complement Technology (TX, US), and recombinant CFH and CFHR proteins were expressed in HEK cells. More specifically, anti-CFH antibodies demonstrated no cross-reactivity to CFHR proteins as measured by ELISA, whereas anti-CFHR1 antibodies showed trace cross-reactivity to CFHR2, however, with CFHR2 binding signal below the signal level of CFHR1 used to extrapolated unknown plasma levels”

R Fig. 2 The use of anti-FH (CFH) antibodies in an ELISA against CFH and CFHR proteins expressed in HEK cells is highlighted in the left panel, while the use of anti-FHR1 (CFHR1) antibodies in the same system is highlighted in the right panel.

Reviewer #2 (Remarks to the Author):

Emilsson and colleagues performed analysis to associate serum protein levels with different types of AMD in the Ages cohort, followed by mapping genetic variants that related to both serum protein levels and risk of AMD, and finally, used MR to identify AMD causal proteins. They highlighted three proteins, CFHR1, CFHR5, and FUT5 to be causally linked to AMD. This comprehensive study leverages large proteomics data and by mapping them with the genetic architecture of AMD, adds to our understandings of the disease pathobiology. I have a few comments regarding the paper and particularly its presentation, please find them below.

Major comments:

I found the two sections of pathway analysis (lines 136-232) too long and convoluted. From what I understand, the authors map the risk variants of AMD to pQTLs, and many of these AMD risk variants (particularly variants at CFH locus) are highly pleiotropic in serum proteins (ie. Only 6 SNPs accounted for 22 of the 28 protein-AMD associations). And the rationale of section “serum proteins regulated...” is not very clear, it seems like this section is more of a discussion of the previous one.

Response: We agree with the reviewer and, as a result, significant portions of the two Result sections “*Proteins in circulation and their co-regulatory networks elucidate the genetic basis of advanced AMD*” and “*Serum proteins regulated by AMD-associated genetic variants map to core pathways involved in the pathobiology of AMD*” have been moved to a new Supplementary Note 1.

In addition, the two Result sections have been combined into one section titled “*Serum proteins elucidate the genetic basis of advanced AMD and highlight key pathways involved in its pathobiology*”.

Finally, any detailed description of the functions and pathways associated with serum proteins of interest from the Discussion section has been moved to the new Supplementary Note 1.

Figures 2b-e and figure 3 also seem a bit repetitive (ie. the titles for Figures 2 and 3 look like they are showing the same thing), is it possible to merge these figures?

Response: We agree with the reviewer; Fig. 2a is now a separate Fig. 5, and Figs. 2b-e and 3 are now combined into Fig 6 as suggested by the reviewer. The previous Fig. 1 has been replaced by Figs. 1 and 2. With the new data on PRMT3 and predictors of late-stage AMD, two new figures have been created (Fig. 3 and Fig. 4).

Sentences describing the gene/protein functions should be moved to discussion to make the core message clear and easy to read.

Response: We completely agree with the reviewer and, as previously stated, have updated the results sections to reflect this. Overall, we revised the Result sections by relocating any gene function-related points to a new Supplementary Note 1 titled “*Proteins associated with AMD-linked genetic variants map to core pathways involved in AMD pathobiology.*” In addition (see same reviewer's response to comment 1 above), much of the discussion on pathways and functions

related to proteins linked to AMD has been moved to the same Supplementary Note 1 to avoid duplicate text and too many words.

Because the proteins in the same cluster are likely not independent to each other, it is expected that most closely linked proteins (like in PM13) tend to also associate with the same outcome. Methods such as lasso/elastic net could pick independent proteins, which can be an interesting addition to the paper, particularly if the highlighted proteins are selected by lasso.

Response: Proteins in the same cluster (module) are co-regulated (assessed through pair-wise correlation), which means that they vary in the same way across the population. It is correct that they are functionally related and affect similar outcomes; however, proteins in the same modules exhibit varying degrees of association to outcome depending on their topology (e.g., hub proteins are more strongly associated with outcome than non-hub proteins). We appreciate that LASSO is being used to answer a different question than protein networks, and it could be a useful addition to the current study, which we have included in a new section called "*Proteins that indicate the progression of or predict late-stage AMD*" that also includes the new PRMT3 progression biomarker results.

To identify protein predictors for late-stage AMD, we used a data-driven nonparametric bootstrap (500 iterations) and least absolute shrinkage and selection operator (LASSO) regression analyses. In our analysis, we adjusted the LASSO models for age and sex by leaving their coefficients unpenalized. In this study, we discovered 21 proteins that appear in at least 80% of the bootstrap iterations. In a receiver operating characteristic curve (ROC) analysis, these 21 proteins were used as a classifier, increasing the AUC from 0.77 for age plus sex to 0.86 for proteins plus age and sex (F-test of equality $P = 1.4 \times 10^{-17}$ for comparison of the two ROC curves). To demonstrate this, two new figures were created: Fig. 4 and Supplementary Fig. 8. Among the 21 proteins were CFHR1, CFHR5, FUT5, BIRC2, and NDUFS4 (all in Table 1), as well as CFH and CFB (see Fig. 4 and Supplementary Fig. 8 below).

The following text has been added to the Result section page 7, lines 145 to 157:

“Next, we applied a data-driven nonparametric bootstrap¹⁸ and least absolute shrinkage and selection operator (LASSO)¹⁹ to estimate the sampling distribution of logistic regression coefficients for all 4782 proteins in order to identify independent protein predictors of advanced AMD (Methods). The 21 proteins that appeared in at least 80% of the 500 iterations included the AMD-associated proteins CFHR1, CFHR5, FUT5, BIRC2, and NDUFS4 (Fig. 4). Here, CFHR1 and NDUFS4 had coefficients that were always non-zero in predicting late-stage AMD (Fig. 4). Interestingly, CFH and CFB are among protein predictors not listed in Table 1. Supplementary Fig. 8 depicts a receiver operating characteristic curve (ROC) for the diagnostic ability of the 21 protein predictors to classify late-stage AMD, demonstrating a significant difference between the two ROC curves, that is the demographics versus the demographics plus proteins ROC curves (F-test of equality $P = 1.4 \times 10^{-17}$). Overall, these findings highlight the additional benefit of using LASSO regression to uncover new aspects of the relationship between global serum proteins and AMD.”

Lines 486-494 have also been added to Methods page 22:

“For identification of protein predictors for late-stage AMD, we approximated the sampling distribution of logistic regression coefficients for all 4782 protein variables, we used the

nonparametric bootstrap¹⁸ and least absolute shrinkage and selection operator (LASSO)¹⁹, estimated using the glmnet package for R²⁰. We summarized our results after 500 bootstrap iterations by calculating the coefficients mean, 95% confidence intervals by calculating the 2.5 percent and 97.5 percent quantiles, and the percentage of iterations in which they were non-zero. We adjusted the LASSO models for age and sex in our analysis by leaving their coefficients unpenalized. Here, we fit the logistic regression LASSO model for prevalent late-stage AMD and compare the odds of late AMD to no AMD”

And to the Discussion section page 13, lines 287-289:

“Some of these proteins, such as CFHR1, CFHR5, BIRC2, and NDUFS4, were found to be among the 21 independent predictors of late-stage AMD”

Fig. 4. Identifying proteins in circulation that predict late-stage AMD. The figure depicts protein predictors for late-stage AMD using a data-driven nonparametric bootstrap and LASSO regression analysis (see Methods for more detail). On the left, we show 21 protein variables that appear in at least 80% of iterations, while the mean estimates and 95% confidence intervals (CIs) are shown on the right. The percentage of iterations with non-zero coefficients for the protein variables is shown in parenthesis.

Supplementary Fig. 8. The figure depicts a receiver operating characteristic curve (ROC) for the diagnostic ability of the 21 protein predictors to classify late-stage AMD. A F-test of equality demonstrates a significant ($P = 1.4 \times 10^{-17}$, two-sided) difference between the two ROC curves, that is the demographics (age + sex, red broken curve) versus the demographics (age + sex) plus proteins (blue curve) ROC curves.

12 of the 28 proteins are in PM13, which modules the rest of the proteins belong to? Is the Eigenprotein for PM13 also associated with other AMD outcomes? What about Eigenproteins for other modules?

Response: We agree and have made a new version of Supplementary Table S4 (now called Supplementary Data 4) to account for the all 28 AMD-associated proteins and the modules they reside in. An updated Fig. 2d highlights the association of Eigenprotein for PM13 to different outcomes showing links to both early AMD and nAMD but not GA AMD. The PM13 and modules in the same supercluster as defined in Emilsson et al.²¹ are the only modules showing significant association to AMD. Given the strong enrichment of the AMD-associated proteins in PM13, we have given a specific attention to that cluster throughout the paper. We should note that in a previous version of Fig. 2, we stated that PM13 is associated with nAMD in the follow-up study, and this remains true, but only if we use early AMD among the controls (for example, excluding early-stage AMD results in only 16 patients with nAMD in the five-year follow-up). However, to ensure consistency throughout the study (see PRMT3 as an example), we would like to use early AMD at baseline and late-stage AMD 5 years later for studies of progression.

For MR, based on the information in table 2, it is easy to guess that the three proteins with cis-SNPs that are also AMD risk variants are the proteins influencing AMD risk (That's why we usually do MR before the association analysis). However, the pleiotropic nature of these SNPs should also be discussed, particularly, the CFH locus SNPs are cis to multiple proteins which are all associated to AMD. In addition, the authors should perform colocalization analysis to exclude possibilities that MR results are confounded by LD.

Response: True, these loci on chromosomes 1 and 19 are not only linked to many proteins, but they also contain many independent loci linked to AMD (Supplementary Data 6). At the CFH and FUT5 loci, there are eight and three independent AMD-linked loci, respectively. Furthermore, there are numerous independent *cis*-acting variants underlying the expression of the proteins

CFHR1, CFHR5 and FUT5^{22,23}. A new Fig. 7b–c (also shown below) depicts scatter plots with the inverse variance weighted causal estimate and MR-Egger regression for CFHR1, CFHR5, and FUT5, which were all identified as causal candidates in the MR analysis. The P-values for the Egger intercept were greater than 0.05, indicating that pleiotropy was not apparent, but this value was near the limit for CFHR5 (Fig. 7b-d).

This new information is highlighted on page 11, lines 235-239 in the Result section:

“Figures 7b–c show scatter plots with the inverse variance weighted causal estimate and MR-Egger regression for CFHR1, CFHR5, and FUT5, which were all identified as causal candidates in the MR analysis. The P-values for the Egger intercept were greater than 0.05, indicating that there wasn't statistically significant evidence of pleiotropy, but this value was near the limit for CFHR5 (Fig. 7b-d).”

As previously stated, these loci contain multiple independent signals for both the proteins^{22,23} as well as the disease (Supplementary Data 6). Colocalization analysis on such loci is notoriously difficult^{24,25}, if not impossible. Under these conditions, we cannot assume that the two traits (protein vs. AMD) share a single causal variant. However, we experimented with various colocalization strategies. First the SuSiE^{25,26} paradigm was used to relax the single-variant assumption by allowing for multiple causal variants. The SuSiE framework requires knowledge of the variants' LD structure. Because the AMD data's LD structure was unavailable, the AGES data's LD structure was used for both sets of variants. The coloc.susie function allows us to conduct colocalization analysis within the SuSiE framework. The coloc.susie function converged successfully for protein FUT5, but not for CFHR1 and CFHR5, which map to the same region. To treat each protein uniformly a second ad-hoc approach was designed: 1. Use the finemap.signals function with the conditional option activated to detect up to 10 independent signals ($P < 5 \times 10^{-8}$) for AMD. 2. Use the finemap.signals function with the conditional option activated to detect up to 10 independent signals ($P < 5 \times 10^{-8}$) for the protein. 3. The LD between the variants belonging to the two sets was computed. The LD-matrix for the lead variants obtained by the finemap.signals function, as well as a scatter plot, are included for each protein (R Figs. 2-4, below). The AMD variants are represented by the rows of the LD matrix, while the protein variants are represented by the columns. The scatter plots' x-axis represents the $-\log_{10}$ transformation of the signal's P-value in the protein data. Similarly, in the AMD data, the y-axis represents the $-\log_{10}$ transformation of the signal's P-value. If the points belonged to either set, as defined in steps 1 and 2 of the third approach, they were colored. Variants which belonged to both sets or were in high LD ($r^2 > 0.8$) were colocalized. This method produced four colocalized signals for CHFR1 (R Fig. 3); one variant was shared by both traits, and three variants had a high level of similarity. For CFHR5 (R Fig. 4) and FUT5 (R Fig. 5), no variants were shared or in high LD. However, as illustrated by the scatterplot for FUT5 (R Fig. 5), there appear to be two distinct signals in this region for the FUT5 protein, though only one is detected by the finemap.signals function, the other being the lead signal for AMD, implying colocalization. We have highlighted one possible explanation for the difficulty in performing the colocalization analysis at these loci, which is that all three proteins are in a region of the genome that is dense with independent signals for both AMD and the proteins:

Lines 239-242 in the Result section on page 11:

“When a locus contains multiple independent variants underlying a trait, colocalization analysis becomes difficult^{24,25}. We attempted colocalization analyses, but because these loci are saturated

with multiple independent variants for AMD (Supplementary Data 6) as well as for the proteins^{22,23}, the results were inconclusive (data not shown).”

Fig. 7. A two-sample MR analysis of the 28 AMD-associated proteins. **a** The causal estimate (red squares) from the two-sample MR analysis compared to the observational estimates (circles) for each of the five proteins with *cis*-acting instruments and associated with AMD in the observational study. As each protein could have different observational coefficients depending on which definition of early AMD (see Methods) was used, it was decided to select and display the coefficient for each definition which had the lower adjusted P-value. The direction of the causal estimate and observational estimates were consistent for proteins CFHR1, CFHR5 and FUT5 and inconsistent for proteins GHR and BPIFB1. The causal estimator for CFHR1, CFHR5 and FUT5 was significant ($FDR < 0.05$) and positive, indicating that an increase in the serum level of these proteins increase the risk of developing AMD. **b** Scatterplot for the CFHR1 protein supported as having a causal effect on AMD in a two-sample MR analysis. The figure demonstrates the estimated effects of the respective *cis*-acting genetic instruments on the serum CFHR1 levels in AGES-RS (x-axis) and risk of AMD through a GWAS provided by the IAMDCG consortium¹⁰ (y-axis). Each data point displays the estimated effect as beta coefficient = $\log(\text{odds ratio})$, along with 95% confidence intervals for the SNP effect on disease (vertical lines) or SNP effect on the protein (horizontal lines). The solid line indicates the inverse variance weighted causal estimate (GWLS; $\beta = 0.590$, $SE = 0.108$, $P = 3.7 \times 10^{-6}$, two-

sided), while the dotted line shows the MR-Egger regression. Similar plots are shown for **c** CFHR5 (GWLS; $\beta = 1.175$, SE = 0.356, P = 0.003), and **d** FUT5 (GWLS; $\beta = 0.107$, SE = 0.029, P = 0.00089). The P-values for the Egger intercept and GWLS are displayed at the top of each scatter plot.

R Fig. 3. CFHR1 colocalization analysis

R Fig. 4. CFHR5 colocalization analysis

R Fig. 5. FUT5 colocalization analysis

It should also be interesting to include the protein-AMD association results for 21 MR significant proteins. If only looking at 20+ proteins instead of 4000+, some of them might survive multiple correction.

Response: This is a good suggestion, and as a result, we've added a new supplementary table (Supplementary Data 15). In this study, eight protein-to-AMD associations were nominally significant, however, only one (APOM) survived the Bonferroni adjusted multiple test correction. However, the association between APOM and AMD was directionally inconsistent with the causal test. We have added a sentence to highlight this on page 11, lines 244-248:

“In this study, 21 additional proteins, including ADAM19, C3, CFI, AIF1, and VTN, were found to have a significant causal estimate for AMD (Supplementary Fig. 11). However, none of these proteins were found to be significantly associated with AMD outcomes in this population, or to be

directionally consistent with the causal estimate after adjusting for multiple testing (Supplementary Data 15).”

Minor comments:

Please use OR and 95% CI for logistic regression results.

Response: In fact, for one unit change in predictor variables, all β -coefficients from the logistic regression analyses are $\log(\text{OR})$. To put it another way, $\text{OR} = \exp(\beta\text{-coefficient})$. There are compelling reasons to use β -values rather than OR, including easier comparisons to studies we and others have published, as well as for possible meta-analyses of the results, given that meta-analyses are based on the β -values. As a result, for the logistic regression analyses, the column headers of tables in the main text and supplementary material have been renamed $\beta = \log(\text{OR})$. We hope the reviewer appreciates our firm stance on using beta values instead of OR values in our proteogenomic analyses.

Table S12: I expect to see the association of 52 AMD risk variants with 4000+ proteins, why the table seems to only have 4 variants?

Response: We apologize for the ambiguity here. In fact, the association of all 52 independent GWAS AMD variants to the 4782 individual proteins was investigated, and these variants, along with the number of serum proteins associated with them, are highlighted in Supplementary Table S6 (now named Supplementary Data 6). Supplementary Table S12 (now called Supplementary Data 12), on the other hand, emphasizes the significance of these variants' associations with each serum protein network (i.e., to their Eigenproteins), emphasizing only those that were significantly associated. Here we used Bonferroni adjustment where associations at $P\text{-value} < 1 \times 10^{-6}$ were considered significant. We have included text in the legends of these tables to help clarify this. In addition, we used the opportunity to review all other tables for clarity and greater consistency.

In line 198-199, which are the “previously identified pathways and new pathways”? It’s better to include those information in a table (or supplementary table), because it’s not very clear in the text that follows.

Response: In this context, we should have cited Supplementary Table S9 (now called Supplementary Data 9), which we have now done (see below). In addition (see response to previous comment), much of the discussion in the current paper on pathways and functions related to proteins linked to AMD has been moved to a new Supplementary Note 1.

The text has been changed at page 9, lines 193-194:

“The proteins linked to AMD-related genetic variants map to pathways with both known and previous unknown association to the disease (Supplementary Note 1, Supplementary Data 9)”

Line 160-161: Possible citing error for Figure 2b.

Response: We thank the reviewer for pointing this out: given the first edition of the paper, it should be Fig. 2b-e, and we've corrected it. Fig. 2b-e, on the other hand, is now Fig. 6a-d as noted above.

There are still some legends missing from some supplementary tables. Ie. What's AUC in Table S2-3?

Response: We appreciate the reviewer bringing this to our attention. We have now gone over the legends and column headers in all Supplementary Tables (Supplementary Data 1-15) including Tables S2 and S3 (now called Supplementary Data 2 and 3) for clarity (see related responses above, for instance regarding the beta-coefficient). AUC is an abbreviation for area under the curve. We should mention that a pdf of the description of the 15 Supplementary Data tables has been submitted along with this revision.

References

1. Holliday, E.G., *et al.* Insights into the genetic architecture of early stage age-related macular degeneration: a genome-wide association study meta-analysis. *PLoS One* **8**, e53830 (2013).
2. Jonasson, F., *et al.* Five-year incidence, progression, and risk factors for age-related macular degeneration: the age, gene/environment susceptibility study. *Ophthalmology* **121**, 1766-1772 (2014).
3. Joachim, N., *et al.* Incidence and progression of geographic atrophy: observations from a population-based cohort. *Ophthalmology* **120**, 2042-2050 (2013).
4. Perlee, L.T., *et al.* Inclusion of genotype with fundus phenotype improves accuracy of predicting choroidal neovascularization and geographic atrophy. *Ophthalmology* **120**, 1880-1892 (2013).
5. Klein, R.J., *et al.* Complement factor H polymorphism in age-related macular degeneration. *Science* **308**, 385-389 (2005).
6. Haines, J.L., *et al.* Complement factor H variant increases the risk of age-related macular degeneration. *Science* **308**, 419-421 (2005).
7. Edwards, A.O., *et al.* Complement factor H polymorphism and age-related macular degeneration. *Science* **308**, 421-424 (2005).
8. Hageman, G.S., *et al.* A common haplotype in the complement regulatory gene factor H (HF1/CFH) predisposes individuals to age-related macular degeneration. *Proc Natl Acad Sci U S A* **102**, 7227-7232 (2005).
9. Hughes, A.E., *et al.* A common CFH haplotype, with deletion of CFHR1 and CFHR3, is associated with lower risk of age-related macular degeneration. *Nat Genet* **38**, 1173-1177 (2006).
10. Fritsche, L.G., *et al.* A large genome-wide association study of age-related macular degeneration highlights contributions of rare and common variants. *Nat Genet* **48**, 134-143 (2016).
11. Lorés-Motta, L., *et al.* Common haplotypes at the CFH locus and low-frequency variants in CFHR2 and CFHR5 associate with systemic FHR concentrations and age-related macular degeneration. *Am J Hum Genet* **108**, 1367-1384 (2021).
12. Kortvely, E., Hauck, S.M., Behler, J., Ho, N. & Ueffing, M. The unconventional secretion of ARMS2. *Hum Mol Genet* **25**, 3143-3151 (2016).
13. Keilhauer, C.N., Fritsche, L.G., Guthoff, R., Haubitz, I. & Weber, B.H. Age-related macular degeneration and coronary heart disease: evaluation of genetic and environmental associations. *Eur J Med Genet* **56**, 72-79 (2013).
14. Skerka, C., *et al.* Defective complement control of factor H (Y402H) and FHL-1 in age-related macular degeneration. *Mol Immunol* **44**, 3398-3406 (2007).
15. Cipriani, V., *et al.* Beyond factor H: The impact of genetic-risk variants for age-related macular degeneration on circulating factor-H-like 1 and factor-H-related protein concentrations. *Am J Hum Genet* **108**, 1385-1400 (2021).
16. Heesterbeek, T.J., *et al.* Complement Activation Levels Are Related to Disease Stage in AMD. *Invest Ophthalmol Vis Sci* **61**, 18 (2020).
17. Cipriani, V., *et al.* Increased circulating levels of Factor H-Related Protein 4 are strongly associated with age-related macular degeneration. *Nat Commun* **11**, 778 (2020).
18. Efron, B. Bootstrap Methods: Another Look at the Jackknife. *The Annals of Statistics* **7**, 1-26, 26 (1979).
19. Tibshirani, R. Regression Shrinkage and Selection Via the Lasso. *Journal of the Royal Statistical Society: Series B (Methodological)* **58**, 267-288 (1996).
20. Friedman, J., Hastie, T. & Tibshirani, R. Regularization Paths for Generalized Linear Models via Coordinate Descent. *J Stat Softw* **33**, 1-22 (2010).
21. Emilsson, V., *et al.* Co-regulatory networks of human serum proteins link genetics to disease. *Science* **361**, 769-773 (2018).
22. Gudjonsson, A., *et al.* A genome-wide association study of serum proteins reveals shared loci with common diseases. *Nature Communications* **13**, 1-13 (2022).

23. Emilsson, V., *et al.* Coding and regulatory variants are associated with serum protein levels and disease. *Nature Communications* **13**, 1-11 (2022).
24. Giambartolomei, C., *et al.* Bayesian test for colocalisation between pairs of genetic association studies using summary statistics. *PLoS Genet* **10**, e1004383 (2014).
25. Wallace, C. A more accurate method for colocalisation analysis allowing for multiple causal variants. *PLoS Genet* **17**, e1009440 (2021).
26. Wang, G., Sarkar, A., Carbonetto, P. & Stephens, M. A simple new approach to variable selection in regression, with application to genetic fine-mapping. *bioRxiv*, 501114 (2020).

REVIEWER COMMENTS

Reviewer #1 (Remarks to the Author):

No further comments.

Reviewer #2 (Remarks to the Author):

The authors have addressed all my previous comments satisfactorily. I have a few minor additional comments.

Line 121: "...we examined which if any of the 4137 proteins...while still in early AMD..." This sentence is a bit confusing, did the authors wish to express that proteins measured in patients with early AMD predicts the risk to late AMD? Please clarify.

Line 185: Paragraph of SLC5A8 looks a bit abrupt. It is not one of the 28 proteins or mentioned in Table 2 or Fig 5 of the preceding paragraph, it would be helpful to describe at the beginning of this paragraph why it is specifically prioritized.

What is the result of the single cell RNA sequencing? It seems that it is only briefly mentioned in Line 117-119. Are the highlighted proteins associated with different stages of AMD or progression of AMD (PRMT3) have support from single cell RNA sequencing result? Same with ELISA of CFHR1, this replication deserves a separate paragraph (Lines 231-233).

CFHR1/CFHR4/CFHR5 appear to be located in the same cluster and are the risk proteins for AMD, which also share several cis-pQTLs. Why CFHR4 is not tested by MR? Figure 6 shows it has a cis-pQTL and also associated with early AMD. It would also be interesting to discuss the biological similarities and differences of CFHR4 vs CFHR5, as it appears that CFHR4 is associated with late AMD while CFHR5 associated with early AMD, with independent cis-pQTLs in gene CFH. entary tables. Ie. What's AUC in Table S2-3?

Response to Reviewers

We are pleased to submit our revised manuscript entitled “*A Proteogenomic Signature of Age-related Macular Degeneration in Blood*” (NCOMMS-21-31042B) for consideration to be published in *Nature Communications*. The responses to reviewer 2's additional comments are provided below in blue font. Text added to the revised manuscript has been italicized. We also took advantage of the opportunity to correct a few typos in the paper and have included precise P-values in Supplementary Data 7, 8 and 13 if a P-value is zero. Page and paragraph numbers listed below refer to the position of the text in the *clean* version of the revised manuscript (submitted along with a manuscript text file highlighting all changes using the track changes mode).

Reviewer #2 (Remarks to the Author):

The authors have addressed all my previous comments satisfactorily. I have a few minor additional comments.

Response: We appreciate the reviewer's positive feedback on the manuscript changes we made.

Line 121: “...we examined which if any of the 4137 proteins...while still in early AMD...” This sentence is a bit confusing, did the authors wish to express that proteins measured in patients with early AMD predicts the risk to late AMD? Please clarify.

Response: We apologize for the confusion and have suggested the following text change to improve clarity.

Page 6, lines 120 to 123:

“Using single point sex and age-adjusted logistic regression analysis, we examined which, if any, of the 4137 serum proteins in early AMD subjects only anticipated advancement to late AMD (pure GA or nAMD) in the same people over a 5-year follow-up period.”

Line 185: Paragraph of SLC5A8 looks a bit abrupt. It is not one of the 28 proteins or mentioned in Table 2 or Fig 5 of the preceding paragraph, it would be helpful to describe at the beginning of this paragraph why it is specifically prioritized.

Response: We thank the reviewer for bringing this to our attention, and we have added the following sentence at the beginning of the paragraph as a result.

Page 9, lines 185 to 187:

“Aside from the expected enrichment of the complement system among the 340 proteins associated with AMD-linked variants (Supplemental data 7 and 9), there were many novel links, including SLC5A8 (aka SMCT1).”

What is the result of the single cell RNA sequencing? It seems that it is only briefly mentioned in Line 117-119. Are the highlighted proteins associated with different stages of AMD or

progression of AMD (PRMT3) have support from single cell RNA sequencing result? Same with ELISA of CFHR1, this replication deserves a separate paragraph (Lines 231-233).

Response: Two separate single-cell RNA-sequencing experiments on seven human donor eyes yielded the data for single cell gene expression¹, also found at the Gene Expression Omnibus (GEO) database (accession nr. GSE135922). The first study looked at single cells from the RPE/choroid (2 controls and 1 AMD), whereas the second looked at the endothelial population after a CD31 antibody enrichment step (3 controls and 1 AMD)¹. Each of the AMD donors was labeled as having "neovascular AMD," with no other phenotypic information provided. Due to the study's small size and methodological limitations, we were not able to assess different stages of AMD or progression. Consequently, we only reported cell-specific expression to assist future researchers in developing hypotheses about the ocular expression of the AMD-associated proteins identified in serum.

To better highlight these details, we have updated the Method section description "*Analysis of gene expression in single cell RNA sequencing data from eye tissues.*"

Page 21, lines 456-470

“Analysis of gene expression in single cell RNA sequencing data from eye tissues: Two separate single-cell RNA-sequencing experiments on seven human donor eyes, five controls and two AMD patients, yielded the data for single cell gene expression¹, also found at the gene Ex-pression Omnibus (GEO) database (accession nr. GSE135922). The first study looked at single cells from the RPE/choroid (2 controls and 1 AMD), whereas the second looked at the endothelial population after a CD31 antibody enrichment step (3 controls and 1 AMD)¹. Each of the AMD donors was labeled as having "neovascular AMD," with no other phenotypic information provided. The normalized single cell data was downloaded from GEO and was analyzed with the R package Seurat (v.3.0.0) in R 3.6.3 environment. The final dataset contained 4335 cells after filtering. Variable genes were identified using Seurat with default parameters and Principal Components Analysis (PCA) was performed on these variable genes. First 11 PCs of the single cell data (resolution = 0.2) were used for clustering cells with similar gene expression profile. Clusters were identified using FindNeighbors and FindClusters functions from Seurat package and UMAP dimensionality reduction was utilized for cluster visualization. The cell clusters were then manually annotated based on the markers reported in the paper¹”

We respectfully disagree with the suggestion that the 3-line summary of the ELISA CFHR1 result be presented as a separate paragraph. Given the weight of many other data and points highlighted and for the sake of clarity and brevity, we do not believe this deserves to be presented in a separate paragraph.

CFHR1/CFHR4/CFHR5 appear to be located in the same cluster and are the risk proteins for AMD, which also share several cis-pQTLs. Why CFHR4 is not tested by MR? Figure 6 shows it has a cis-pQTL and also associated with early AMD. It would also be interesting to discuss the biological similarities and differences of CFHR4 vs CFHR5, as it appears that CFHR4 is associated with late AMD while CFHR5 associated with early AMD, with independent cis-pQTLs in gene CFH.

Response: These genes are colocalized and share *cis*-acting pQTL instruments, which is correct. When multiple comparisons were made, CFHR4 was not found to be one of the proteins with the strongest links to AMD (Table 1). In other words, because there was no observational link between CFHR4 and AMD at the study-wide significant P-value threshold, the protein was not directly tested for a causal relationship with AMD using MR analyses. When assessing the relationship (causal or reverse causation) between exposure (proteins in this case) and outcome, we, like many others, begin with the observed link between exposure and outcome. The next step is to locate suitable genetic instruments. It is worth noting that when we included all 1327 *cis*-acting genetic instruments (including CFHR4) regardless of any previously observed link of a protein to AMD, 21 additional proteins show a significant causal estimate for AMD using a more stringent multiple testing correction threshold. CFHR4, however, was not among these proteins (Supplementary Fig. 11). In fact, the uncorrected P-value for CFHR4 in this case was = 0.09, while the adjusted FDR was 0.40. When CFHR4 was examined in a single test with quintile comparisons, it was found to be weakly but significantly associated with early-stage AMD. In summary, our findings do not rule out the possibility of a causal relationship between CFHR4 and AMD, but with the current sample size and multiple testing corrections, it was not among those with a significant causal estimate for AMD.

The implications of the various CFHR proteins associated with different AMD stages and AMD variants are currently unknown. Both CFHR proteins linked to AMD (CFHR1 and CFHR5) belong to the Group I CFHR subgroup that circulate as dimers². Group II family members CFHR4 and CFH, on the other hand, lack the N-terminal dimerization domains², and are not strongly associated with AMD despite both associated with AMD variants. It is unclear what role, if any, the distinct structural features of these proteins play here. In passing, we should mention that the aptamer binding specificity for CFH, CFHR1, CFHR4, and CFHR5 have all been validated using pull-down and mass spectrometry analyses (Supplementary Data 14 and ref. Emilsson et al.³).

References

1. Voigt, A.P., *et al.* Single-cell transcriptomics of the human retinal pigment epithelium and choroid in health and macular degeneration. *Proc Natl Acad Sci U S A* **116**, 24100-24107 (2019).
2. Skerka, C., Chen, Q., Fremeaux-Bacchi, V. & Roumenina, L.T. Complement factor H related proteins (CFHRs). *Mol Immunol* **56**, 170-180 (2013).
3. Emilsson, V., *et al.* Co-regulatory networks of human serum proteins link genetics to disease. *Science* **361**, 769-773 (2018).

REVIEWERS' COMMENTS

Reviewer #2 (Remarks to the Author):

The authors addressed all my questions and I have no additional comments